



# The impact and estimation of uncertainty correlation for multi-angle polarimetric remote sensing of aerosols and ocean color

Meng Gao[1,2], Kirk Knobelspiesse[1], Bryan A. Franz[1], Peng-Wang Zhai[3], Brian Cairns[4], Xiaoguang Xu[3], and J. Vanderlei Martins[3]

[1]NASA Goddard Space Flight Center, Code 616, Greenbelt, Maryland 20771, USA
[2]Science Systems and Applications, Inc., Greenbelt, MD, USA
[3]JCET/Physics Department, University of Maryland, Baltimore County, Baltimore, MD 21250, USA
[4]NASA Goddard Institute for Space Studies, New York, NY 10025, USA

**Correspondence:** Meng Gao (meng.gao@nasa.gov)

**Abstract.** Multi-angle polarimetric (MAP) measurements contain rich information for characterization of aerosol microphysical and optical properties that can be used to improve atmospheric correction in ocean color remote sensing. Advanced retrieval algorithms have been developed to obtain multiple geophysical parameters in the atmosphere-ocean system, although uncertainty correlation among measurements is generally ignored due to lack of knowledge on its strength and characterization. In this work, we provide a practical framework to evaluate the impact of the angular uncertainty correlation from retrieval results and a method to estimate correlation strength from retrieval fitting residuals. The Fast Multi-Angular Polarimetric Ocean coLor (FastMAPOL) retrieval algorithm, based on neural network forward models, is used to conduct the retrievals and uncertainty quantification. In addition, we also discuss a flexible approach to include a correlated uncertainty model in the retrieval algorithm. The impact of angular correlation on retrieval uncertainties is discussed based on synthetic AirHARP and HARP2 measurements using a Monte Carlo uncertainty estimation method. Correlation properties are estimated using auto-correlation functions based on the fitting residuals from both synthetic AirHARP and HARP2 data and real AirHARP measurement, with the resulting angular correlation parameters found to be larger than 0.9 and 0.8 for reflectance and DoLP, respectively, which correspond to correlation angles of $10°$ and $5°$. Although this study focuses on angular correlation from HARP instruments, the methodology to study and quantify uncertainty correlation is also applicable to other instruments with angular, spectral, or spatial correlations, and can help inform laboratory calibration and characterization of the instrument uncertainty structure.

## 1 Introduction

Satellite remote sensing is important for the study of the earth system at a global scale (National Academies of Sciences, Engineering, and Medicine, 2018). Remote sensing instruments are evolving rapidly, with increasing accuracy and spatial, spectral, and angular resolutions (Kokhanovsky et al., 2015; Dubovik et al., 2019). Multi-angle polarimeters (MAPs), measuring polarization states at multiple spectral bands and viewing angles, contain high information content for the study of aerosol and cloud optical and microphysical properties (Mishchenko and Travis, 1997; Chowdhary et al., 2001; Hasekamp and Landgraf, 2007;



Knobelspiesse et al., 2012). The aerosol properties derived from MAP instruments can be used to assist atmospheric correction for ocean color remote sensing (Frouin et al., 2019; Gao et al., 2020; Hannadige et al., 2021).

Uncertainty quantification from MAP retrievals provides information on the quality of the data products and improves our understanding of retrieval sensitivities. These uncertainties depend on the retrieval algorithm as well as the instrument characterization, including the spectral bands, viewing angles, and polarization capability, and the measurement accuracy. As shown in Fig. 1 and table 1, MAP instruments collect a large number of high quality measurements with differing numbers of spectral bands and viewing angles (Gao et al., 2021b). The number of spectral bands are mostly within 4-14 for MAP instruments. An exception is the SPEX sensors (SPEX airborne and SPEXone, Smit et al. (2019); Hasekamp et al. (2019)), which acquire up to 400 spectral bands. The number of viewing angles for many instruments vary between 5 and 16 (e.g., 5 for SPEXone, 9 for SPEX airborne and MISR), but can be on the order of 100 for several hyper-angular instruments (90 for HARP2, 120 for AirHARP, 152 for RSP and 250 for APS). More viewing angles are preferred for the observation of clouds (Waquet et al., 2009; McBride et al., 2020), and they can also be used to conduct multi-angle cloud masking and data screening to increase aerosol retrieval accuracy and coverage (Gao et al., 2021b).

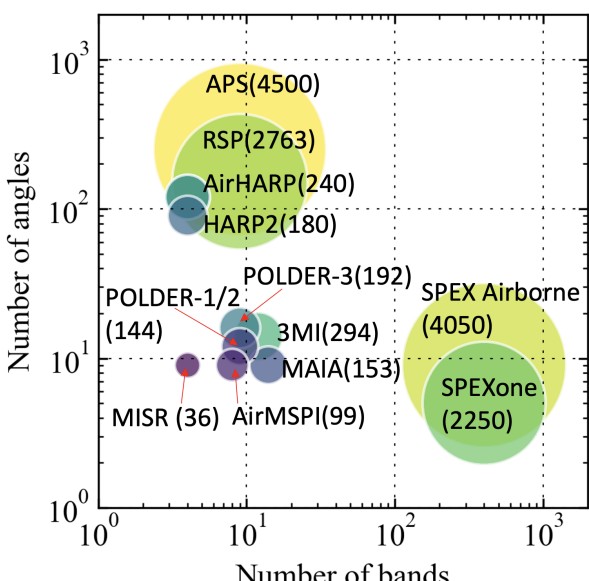

**Figure 1.** Current MAPs in terms of the number of spectral bands and total number of viewing angles as summarized in Gao et al. (2021b). The bubble size for each instrument corresponds to the total number of measurements as indicated next to the instrument name. The full name and reference of the instruments are provided in Table 1. POLDER-1, -2, and -3 refer to the instruments on ADEOS-I, ADEOS-II and PARASOL missions. Note that MISR conduct multi-angle measurement without considering polarization.

To understand the retrieval uncertainties, an uncertainty model is required to describe the combined uncertainties from the MAP measurements, forward model, and a priori assumptions. These combined uncertainty sources are often assumed to be independent and without correlations; however, measurements with high angular or spectral resolution are likely to have



**Table 1.** Acronyms and their definitions for the MAP instruments plotted in Fig. 1. The number of angle and spectral bands are summarized in Gao et al. (2021b). Details in the instrument characteristics are available in Dubovik et al. (2019).

| Instruments | Full name | Reference |
|---|---|---|
| POLDER/ADEOS I and II | Polarization and Directionality of the Earth's Reflectances on Advanced Earth Observing Satellite missions I and II | Deschamps et al. (1994) |
| POLDER/PARASOL | POLDER on Polarization and Anisotropy of Reflectances for Atmospheric Sciences coupled with Observations from a Lidar mission | Tanré et al. (2011) |
| 3MI/MetOp-SG | Multi-Viewing Multi-Channel Multi-Polarisation Imaging instrument on Meteorological Operational - Second Generation mission | Fougnie et al. (2018) |
| MISR/Terra | Multi-angle Imaging SpectroRadiometer on Terra mission | Diner et al. (1998) |
| AirMSPI | Airborne Multiangle SpectroPolarimetric Imager | Diner et al. (2013) |
| MAIA | Multi-Angle Imager for Aerosols | Diner et al. (2018)) |
| RSP | Research Scanning Polarimeter | Cairns et al. (1999) |
| APS/Glory | Aerosol Polarimetry Sensor on Glory mission | Mishchenko et al. (2007) |
| AirHARP | Airborne Hyper-Angular Rainbow Polarimeter | Martins et al. (2018) |
| HARP2/PACE | Space-borne version of AirHARP on PACE mission | Martins et al. (2018) |
| SPEX Airborne | Spectro-Polarimeter for Planetary EXploration Airborne | Smit et al. (2019) |
| SPEXone/PACE | Space-borne version of SPEX on PACE mission | Hasekamp et al. (2019) |

correlated uncertainty, depending on instrument design. For example, a sensor may use the same detector to scan through all measurement view angles (e.g. the Research Scanning Polarimeter, RSP, Cairns et al. (1999)), and thus the systematic errors due to calibration will be correlated for different measurement angles. The correlation property should be part of the MAP uncertainty model, but often it has not been sufficiently characterized. The characterization of measurement uncertainty

correlation is affected by instrument calibration and data processing protocols, which is challenging to quantify. A sensitivity study considering the angular uncertainty correlation of the RSP data was conducted by Knobelspiesse et al. (2012), which showed that information content is affected by correlation strength. However, the actual impacts of uncertainty correlation in a retrieval algorithm have not been well explored, as this requires better understanding of the correlation characteristics and efficient implementation in a retrieval algorithm.

Retrieval algorithms that exploit correlation information in retrieval parameters and measurement uncertainties have shown benefits in improving remote sensing capabilities. The Generalized Retrieval of Aerosol and Surface Properties (GRASP) algorithm retrieves multiple pixels simultaneously, while considering the spatial correlation of the retrieval parameters (Dubovik et al., 2014, 2021). Xu et al. (2019) developed a correlated multi-pixel inversion approach (CIMAP), which further considers the correlation between different retrieval parameters. Theys et al. (2021) developed a Covariance-Based Retrieval Algorithm

(COBRA) based on an error covariance matrix estimated from measurements with spectral correlation, applied their approach



to sulfur dioxide (SO2) retrievals from the TROPOspheric Monitoring Instrument (TROPOMI) data, and demonstrated improved retrieval performance.

In this study, we provide a practical framework to understand the measurement uncertainty structure, study the impact of correlation in MAP retrievals, and demonstrate the potential for improvement in geophysical retrieval performance when proper

correlation information is incorporated into the retrieval algorithm. Angular uncertainty correlation in measurements from the AirHARP and HARP2 instruments are studied as examples. Both instruments measure 60 angles at 670nm. AirHARP measures 20 angles for the 440, 550, and 870nm bands, while HARP2 measures at 10 angles for these bands. Angular correlation within each band is considered and modeled separately. Two methods are used to evaluate the retrieval uncertainties under different correlation strengths: 1) the error propagation method is used to evaluate the optimal retrieval uncertainties, by mapping

the input uncertainty model describing the total uncertainty of the measurement and forward model to the retrieval parameter domain, and 2) comparative analyses are performed between the retrieval results from synthetic MAP measurements and the "truth data" that was assumed in the generation of that synthetic MAP data. A Monte Carlo uncertainty estimation method (MCEP) is adopted to compare the retrieval uncertainties from these two methods. To efficiently conduct retrieval and uncertainty analysis, the FastMAPOL retrieval algorithm is employed in this study, which uses neural network forward models

for coupled atmosphere and ocean systems (Gao et al., 2021a). Analytical Jacobian matrices are derived based on the neural network and used to improve the efficiency of the retrieval (Gao et al., 2021b) and uncertainty quantification (Gao et al., 2022). To accurately evaluate the retrieval uncertainties of real measurements, an adaptive data screening approach is employed (Gao et al., 2021b). This ensures that only those measurements that can be sufficiently described by the forward model are used in this study, by avoiding uncharacterized uncertainty contributions due to contamination by cirrus clouds and other anomalies.

Furthermore, we study the angular uncertainty correlation in the measurements, and demonstrate that the correlation property can be derived using the autocorrelation function from the retrieval fitting residuals. Studies on both synthetic data with various correlation strengths are conducted with results applied to the real masurement retrievals from AirHARP over multiple ocean scenes. Useful tools are provided to understand and analyze the angular correlated uncertainty structure and models. Note that autocorrelation analysis based on fitting residuals has been found useful in analyzing performance of machine learning

algorithms such as using the Durbin–Watson test (Chatterjee and Simonoff, 2012).

In the following sections, we will discuss how to conveniently include an angular correlated uncertainty model in the retrieval algorithm (Sec. 2), evaluate the impact of correlated measurement uncertainty in retrieval uncertainties (Sec. 3), and estimate correlation strength by fitting residual analysis from both synthetic AirHARP and HARP2 data and real AirHARP measurements (Sec. 4). Discussion and conclusions are provided in Sec. 5. Although this study focuses on angular noise correlation,

the conclusions on the impacts of correlations are also applicable to other instruments such as the hyperspectral measurements from both the SPEXone and the Ocean Color Instrument (OCI) that will be carried on NASA's upcoming Plankton, Aerosol, Cloud, ocean Ecosystem (PACE) mission (Werdell et al., 2019).





## 2 Algorithm and Methodology

### 2.1 FastMAPOL retrieval algorithm

In this study, the FastMAPOL algorithm is used to retrieve aerosol and ocean optical properties from HARP measurements. The algorithm includes three main components: 1) a set of neural network based radiative transfer forward models of the

coupled atmosphere and ocean system (Gao et al., 2021a) and the corresponding analytical Jacobian matrix based on these neural networks (Gao et al., 2021b), 2) a multi-angle cloud masking and data screening module (Gao et al., 2021b), and 3) an efficient uncertainty quantification component (Gao et al., 2022). Water leaving signals in terms of remote sensing reflectance are derived with an additional neural network trained for the atmospheric correction process (Gao et al., 2021a).

The neural network forward models are trained for both reflectance and degree of linear polarization (DoLP) based on

simulations from the successive orders of scattering radiative transfer model (RTSOS) developed by Zhai et al. (2009, 2010, 2022). The atmosphere and ocean system are assumed to be a four-layer system. The bottom layer is the ocean water body in which an open ocean bio-optical model is used to parameterize scattering and absorption of ocean constituents based on the chlorophyll a concentration (Gao et al., 2019, 2021a). The second layer is the ocean surface with its roughness parameterized by wind speed through a scalar Cox-Munk model (Cox and Munk, 1954). The third layer is an aerosol layer mixed with

Rayleigh scattering. The layer extends from ocean surface to a height of 2km. The last layer contains atmospheric molecules from 2 km to the top of the atmosphere.

A total of 9 parameters are used to describe the aerosol microphysical properties. There are four parameters for the complex refractive index of fine and coarse mode. Aerosol size distributions are parametrized by five volume densities for five size submodes with fixed effective radius and variance (Dubovik et al., 2006; Xu et al., 2016; Gao et al., 2018). Absorption by

atmospheric gases is considered in the RTSOS simulation, with ozone density as the only variable. The radiant path geometries are represented by the solar and viewing zenith angles and the viewing azimuth angle relative to the solar direction. Therefore, a total of 15 parameters are used as forward model input, with 11 of them defined as retrievable parameters. Details of the parameter ranges are listed in Appendix A and discussed in Gao et al. (2021a, 2022). To represent the forward model accurately and efficiently, the NN architecture is optimized with an input layer of 15 parameters, followed by three hidden layers with

1024, 256, and 128 nodes and a final output layer with 4 nodes for each HARP band. The deep learning Python library PyTorch is used for the training the NN (Paszke et al., 2019).The accuracy of the NN forward model is examined with an independent synthetic measurement dataset not used in training. An accuracy of less than 1% for reflectance and less than 0.003 for DoLP has been achieved (Gao et al., 2021a).

### 2.2 Retrieval cost function and uncertainty quantification

The maximum likelihood approach is used to retrieve the state parameters in FastMAPOL by minimizing a cost function that represents the difference between the measurements and the forward model fitting (Rodgers, 2000)

$$\chi^2 \quad = \quad \frac{1}{N} \mathbf{y}^T \mathbf{S}_\epsilon{}^{-1} \mathbf{y}, \tag{1}$$



where $\mathbf{y} = \mathbf{m} - \mathbf{f}(\mathbf{x})$ is the residual vector between measurement $\mathbf{m}$ and forward model $\mathbf{f}$ under retrieval parameters of $\mathbf{x}$. Measurement vector $\mathbf{m}$ includes both reflectance ($\rho_t$) and DoLP ($P_t$) with the total number of measurements of $N$, which has been used in previous studies (Gao et al., 2021a).

The error covariance matrix $\mathbf{S}_\epsilon$ in Eq. (1) specifies the uncertainties of each measurement and the correlation between 5 different measurements at the same pixel, which is a symmetric matrix defined as:

$$S_{\epsilon;i,j} = \mathbb{E}[(y_i - \mathbb{E}[y_i])(y_j - \mathbb{E}[y_j])], \tag{2}$$

where $i$ and $j$ indicate the measurement at different angles and bands, and $\mathbb{E}$ indicate the expectation values. To capture the angular uncertainty correlation, the autoregressive model of order of 1 (denoted as AR(1)) is used in the study of RSP data (Knobelspiesse et al., 2012), and adopted in this study for HARP data. AR(1) represents a linear Markov process with error 10 covariance matrix specified as

$$S_{\epsilon,i,j} = \begin{cases} \sigma_{t,i}^2 & \text{if } i = j \\ \sigma_{c,i}\sigma_{c,j}r^{\Delta_\theta|i-j|} & \text{if } i \neq j \text{ but at the same band and polarization state} \\ 0 & \text{otherwise} \end{cases} \tag{3}$$

where $\sigma_t$ is the total uncertainty, which includes both random noise and calibration uncertainty ($\sigma_c$). Only $\sigma_c$ is assumed to be correlated between measurements at different viewing angles.

The ratios between random and calibration uncertainties may be different for reflectance and polarized signals (Knobel-15 spiesse et al., 2019). The synthetic data is generated directly using the forward model, therefore the contribution of forward modeling uncertainty is not considered for the synthetic data study. $\Delta_\theta$ is the average angular grid size, which depends on the channels. We model the correlation properties using the $\Delta_\theta$ estimated from the viewing angles in the along track direction, to better represent the stripe filter characteristics used to conduct HARP angular measurement. The averaged $\Delta_\theta$ is approximately $6.0°$ for AirHARP and $12°$ for HARP2 at 440, 550 and 870 nm bands, and $2.0°$ for the 670 nm bands for both HARP instru-20 ments. $r$ in Eq. (3) is the correlation parameter with a value between 0 and 1. For uncertainties with more complex structures, a general autoregression and moving average (ARMA) model can be used (Priestley, 1983). However from our analysis of AirHARP measurements in Sec. 4.3, AR(1) works well for most cases.

To better represent stronger correlations when it is close to one, we define the correlation angles $\theta_c$ based on the correlation parameter $r$ as

$$r^{\Delta_\theta|i-j|} = e^{-\Delta_\theta|i-j|/\theta_c} \tag{4}$$

Therefore, $\theta_c$ indicates the angular range where magnitude in the correlation between angles is reduced by a factor of $e$. Similarly, correlation angles can be derived from $r$ as

$$\theta_c = -1/\ln r \tag{5}$$





## 2.3 Uncertainty quantification

The pixel-wise retrieval uncertainty can be quantified by mapping the measurement and forward model uncertainties into retrieval parameter space (Rodgers, 2000):

$$\mathbf{S^{-1}} \quad = \quad \mathbf{K^T S_\epsilon^{-1} K + S_a^{-1}} \tag{6}$$

where the Jacobian matrix $\mathbf{K}$ represents the partial derivatives of the measurements with respect to all the retrieval parameters. In this study, each retrieval parameter can only vary in a limited range as shown in Table A1, which imposes an implicit a priori constraint on the retrieval parameters. To capture its influence on retrieval uncertainties, we assume the a priori error matrix $\mathbf{S_a}$ in Eq. (6) to be diagonal with the a prior uncertainty for each state parameter approximated by its permitted range in retrievals (Gao et al., 2022). The uncertainties are defined as the standard deviation ($1\sigma$) around the retrieval solution, which is estimated by the square roots of the diagonal elements of $\mathbf{S}$. The uncertainties of variables which are a function of the retrieval parameters can also be derived from S and their derivatives. Due to the large number of retrieval parameters used in the retrieval, the evaluation of the retrieval uncertainties can be time consuming. The speed to compute uncertainties is improved using automatic differentiations based on neural network forward models (Gao et al., 2022).

The retrieval uncertainties estimated by error propagation (hereafter called theoretical retrieval uncertainty) as shown in Eq. (6) represent the optimal scenarios, with limitations such as the assumption that the retrieval parameters successfully converged to the global minima (more discussions in Sayer et al. (2020); Gao et al. (2022)). To quantify the retrieval uncertainties based on actual retrieval results, the retrieval errors are defined as the difference between the retrieval results and the truth from synthetic data, which are then used to compute the retrieval uncertainty (hereafter called real retrieval uncertainty). The comparison between the theoretical and real uncertainties are useful to access the optimal and actual performance of a retrieval algorithm. The Monte Carlo error propagation (MCEP) method is used in this study to conduct such comparison (Gao et al., 2022). MCEP samples the retrieval errors from theoretical retrieval uncertainties and then directly compares the error distributions between theoretical estimation and real retrievals. This method provides additional flexibility in analyzing their statistics. Multiple sets of random samples are generated from the theoretical uncertainties with their variations analyzed, which therefore provides a way to evaluate the impact of sample size in estimating uncertainties (Gao et al., 2022). This method is used to quantify the retrieval uncertainties with various correlation strength in the next section. A table of the terminology on the error and uncertainties for measurement and retrieval results are provided in Table 2.

## 2.4 Eigenvector decomposition on error covariance matrix

The error covariance matrix with non-diagonal terms is challenging to implement efficiently in optimization algorithms, which typically operate in diagonal space with no correlation between measurements. The error covariance matrix also creates barriers to understand the retrieval uncertainties, as the input uncertainties are not for a single measurement, but rather related to multiple measurements. To overcome these issues, we convert the measurements into a new space where the error covariance matrix is diagonalized. Therefore, conventional optimization and error analysis techniques can be readily used.



**Table 2.** Error and uncertainty definitions.

| Term | Definition |
|------|------------|
| Measurement error | Difference between the real measurement and the physical quantity to be measured |
| Measurement uncertainty | The statistical variation of the measurement errors |
| Retrieval error | Difference between the truth and the retrieval results |
| Retrieval uncertainty | The statistical variation of the retrieval errors around retrieval solution |
| Retrieval fitting residual | Difference between the measurements and the forward model fitting |

To achieve this goal, eigenvector decomposition is applied on the error covariance matrix (Rodgers, 2000) as

$$\mathbf{S}_\epsilon = \mathbf{U^T D}_\epsilon \mathbf{U} \tag{7}$$

where $\mathbf{D}_\epsilon$ is a positive diagonal matrix defined by the eigenvalues of $\mathbf{S}_\epsilon$ and $\mathbf{U}$ is a unitary matrix. Examples of how the eigenvalues vary with correlation strength, and the associated impact of those variations on Shannon information content and retrieval uncertainties are discussed in Appendix B. Based on Eq. (7), the cost function in Eq. (1) and the error propagation in Eq. (6) can be written as

$$\chi^2 = \frac{1}{N}\mathbf{y'}^T \mathbf{D}_\epsilon^{-1} \mathbf{y'}, \tag{8}$$
$$\mathbf{S^{-1}} = \mathbf{K'^T D}_\epsilon^{-1}\mathbf{K' + S_a^{-1}} \tag{9}$$

where the original set of measurements, $\mathbf{y}$, are converted to a new set of measurements, $\mathbf{y'}$, without any correlation, along with the respective Jacobian matrices:

$$\mathbf{y' = Uy}, \tag{10}$$
$$\mathbf{K' = UK} \tag{11}$$

Eqs. (8,9) are mathematical transformations that conveniently allow for working with diagonal matrices, with advantages and applications summarized below:

– A clear expression of measurement uncertainty

  The diagonal terms in matrix $\mathbf{D}_\epsilon$ represent the uncertainties for the new measurement $\mathbf{y'}$, therefore providing insights on the accuracy of the measurements impacted by correlation. An example of this is shown in Appendix B.

– Conducting minimization on retrieval cost function

  Eq. (8) represents the cost function for non-correlated measurement $\mathbf{y'}$, and therefore can be used for conventional optimization algorithms such as the subspace trust-region interior reflective (STIR) algorithm (Branch et al., 1999) as used in current FastMAPOL algorithm (Gao et al., 2021a, b, 2022).

– Generating correlated errors





To study and visualize angular uncertainty correlations, correlated errors need to be generated and then added to the synthetic data. To achieve this goal, we generated the errors in the space for $\mathbf{y}'$ with random parameters sampled from a normal distribution assuming the eigenvalues in $\mathbf{D}_\epsilon$ as its variance. These errors in $\mathbf{y}'$ are then transformed back to the original space $\mathbf{y}$ through $\mathbf{y} = \mathbf{U^T y}'$. The correlated error samples with correlation angle of $\theta_c = 10°$ ($r = 0.9$) and

correlation angle of $\theta_c = 60°$ ($r = 0.98$) are shown in Fig. 2 (a) and (c). With larger $\theta_c$ the errors start to form longer range of correlation with smoother variations. These errors are then added to the synthetic data and used to study retrieval The fitting residual from retrievals on the synthetic data with the added correlated errors are shown in Fig. 2 (c) and (d) and discussed in the next section.

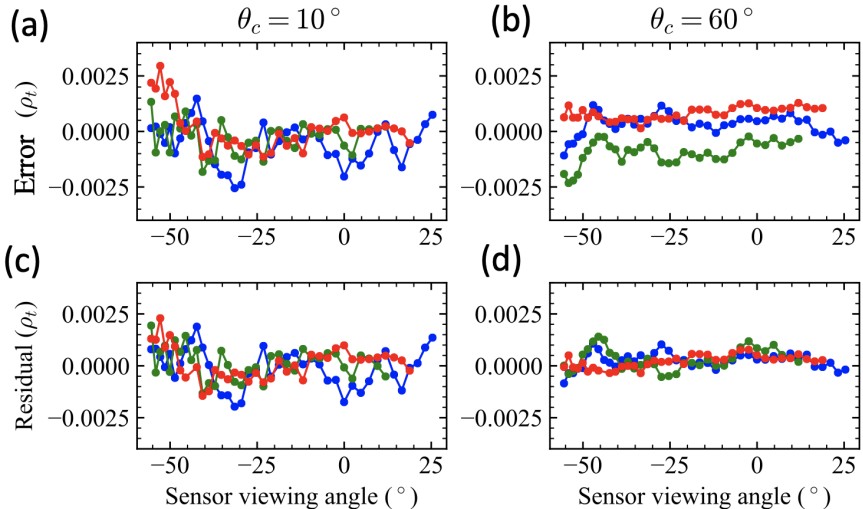

**Figure 2.** Examples of simulated measurement errors generated for reflectance at 660nm band with correlation angle of $10°$ (a) and $60°$ (b), respectively. The fitting residuals are shown in (c) and (d). Three sets of error examples are shown in different colors.

## 2.5 Correlation strength estimation using autocorrelation

Autocorrelation is a useful function to quantify correlation in a discrete data sequence, and is defined as (Priestley, 1983):

$$R_{i,j} \quad = \quad \mathbb{E}[y_i y_j], \tag{12}$$

where $i$ and $j$ are two indices of the datasets. Comparing with Eq.(2), the autocorrelation is equivalent to the autocovariance when $\mathbb{E}[y_i] = 0$. This method can be applied to the simulated noise generated in Sec. 2.4, and used to analyze the fitting residuals. However, the mean values and variance in the fitting residuals often vary with respect to the angular grids. This type

of signal is classified as non-stationary and difficult to model (Priestley, 1983). To overcome this issue, the original residual data $\mathbf{y}$ is processed by removing its mean and normalizing by its standard deviation. This normalized data is denoted as $\tilde{\mathbf{y}}$. For the data within the same band and polarization state, the autocorrelation function on the normalized data is equal to its





covariance as defined in Eq. (3),

$$\tilde{R}_k \quad = \quad \mathbb{E}[\tilde{y}_i \tilde{y}_{i+k}] = r^{\Delta_\theta k}. \tag{13}$$

We can estimate the correlation by analyzing the residuals between the measurement and forward model. The autocorrelation function is averaged over multiple pixels to reduce uncertainties for the analysis in both synthetic data and real retrieval
residuals. The correlation parameter can then be derived as:

$$r = (\tilde{R}_1)^{1/\Delta_\theta} \tag{14}$$

Correlation angles, $\theta_c$, are then computed based on $r$ following the formula in Eq. (5). Furthermore, a partial auto-correlation function from a sequence of data can be computed, which removes the correlation due to lags higher than 1 (Priestley, 1983). If AR(1) model is sufficient to describe the noise structure, only one additional term would be left besides the zero-order term in
the partial autocorrelation results. Therefore, partial autocorrelation can be used to validate our assumption in the noise model. The Python packages, StatsModels (Seabold and Perktold, 2010) and SciPy (Virtanen et al., 2018), are used to conduct the auto-correlation analysis.

An example is shown in Fig. 3, the autocorrelation function and partial autocorrelation function are applied on the simulated errors and the retrieval residuals from Fig. 2 (b) and (d). The autocorrelations are shown in Fig.3 (a) and (c) for the simulated
errors and retrieval residuals in reflectance data. The partial autocorrelations for Fig.3 (a) and (c) are shown in Fig.3 (b) and (d) respectively. For both cases in Fig. 3 (b) and (d), only the first order of data are prominent, which confirms that the data can be represented by the AR(1) process. If higher orders in the AR process are presented, more prominent data point will appear in Fig. 3 (b) and (d). The estimated correlation angles for the errors in Fig. 3 (a,b) and residuals in Fig. 3 (c,d) are approximately $30°$ and $15°$, respectively, after converting correlation parameters to correlation angles but less than the actual
correlation angle of $60°$. The results show that autocorrelation can be a useful way to estimate correlation strength, but with tendency to underestimate due to the finite length of the data and overfitting of the retrievals (more discussion in the Section 4).



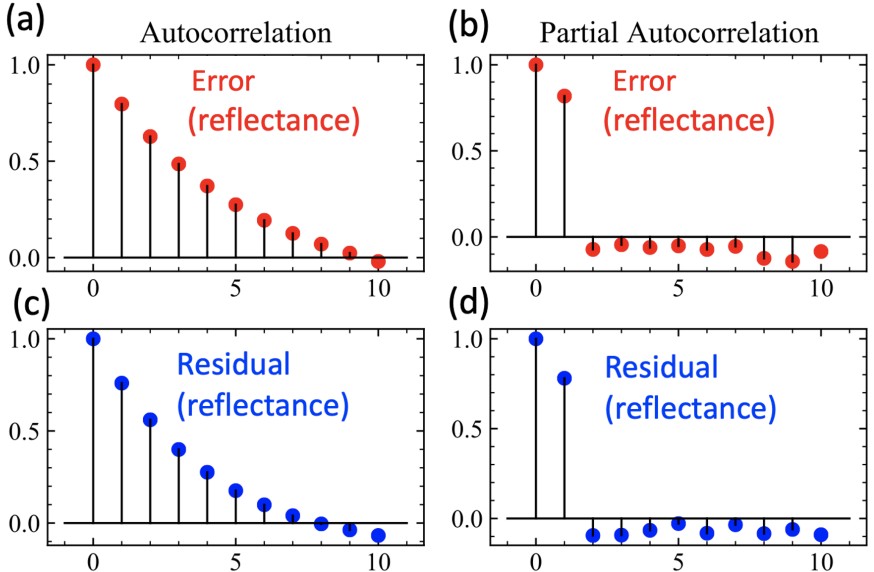

**Figure 3.** The autocorrelation and partial autocorrelation on the simulated errors with correlation angle of $60°$ and its corresponding fitting residuals as shown in Fig. 2 (b) and (d) for 670nm band. The maximum value in the curve is normalized to 1. Horizontal axis indicates the angular step $k$ as defined in Eq. (13).

## 3   Retrieval uncertainties for AirHARP and HARP2

### 3.1   Synthetic data generated using NN forward model

The neural network forward model discussed in Sec. 2.1 is used to generate 1,000 sets of synthetic AirHARP data, and then the number of viewing angles at 440, 550 and 870nm are down sampled to 10 to represent HARP2 data. A fixed solar zenith angle

5 of $50°$ is used to represent the solar geometries of the AirHARP scenes over ocean from the ACEPOL field campaign (more information in Sec. 4.3). The aerosol properties, wind speed, and Chla values are randomly sampled based on their allowed range, as discussed in Sec. 2 and Appendix A. The same sampling approach discussed in Gao et al. (2022) is conducted assuming that the aerosol optical depth (AOD) and fine mode volume fraction are uniformly distributed within [0.01, 0.5] and [0,1], respectively. Realistic HARP-like viewing geometries are constructed by sampling the along-track and cross-track

10 viewing angles randomly and then converting to the actual viewing zenith and azimuth angles following the formulas provided in Gao et al. (2021b). Example viewing geometries for AirHARP and HARP2 for bands at 550nm are provided in Fig. 4, with geometries at other bands constructed similarly.



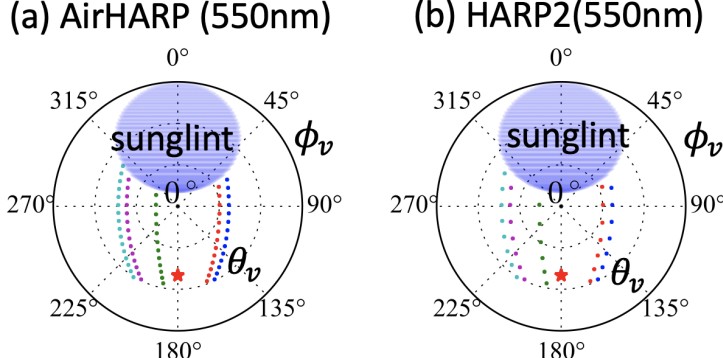

**Figure 4.** Example viewing zenith ($\theta_v$) and relative azimuth ($\phi_v$) in a polar plot at 550nm for AirHARP (20 angles) and HARP2 (10 angles). Five sets of examples are provided in different colored dots. The 440 and 870nm bands are similar. At 670nm, there is a total of 60 angles for both instruments. The anti-solar point is indicated by the red asterisk. The viewing angles within sunglint region indicated by the blue shaded area are removed. Note that the viewing angles from HARP2 are down sampled from AirHARP.

To generate realistic measurement, correlated uncertainties with correlation angle ($\theta_c$) from 0, 1, 2, 5, 10, 20, 30, 60, 120 are considered in this study, with corresponding correlation parameters of 0, 0.368, 0.607, 0.819, 0.905, 0.951, 0.967, 0.983, 0.992 respectively. The corresponding correlated error samples are generated based on the error covariance matrix using the method discussed in Sec. 2.4. Examples of the correlated errors are shown in Fig. 2 for AirHARP at 660 nm with correlation angles of

$10°$ and $60°$ respectively.

Correlated errors for both the AirHARP and HARP2 instruments are generated according to the same 3% uncertainty for reflectance, but 0.01 in DoLP for AirHARP and 0.005 in DoLP for HARP2. These errors are added to the corresponding simulated reflectance and DoLP ($\sqrt{Q^2 + U^2}/I$) for further studies. The reflectance is more likely to be dominated by systematic uncertainty (possibly correlated) like calibration, while DoLP defined as the ratio between two measurements is more likely

dominated by randomly generated uncertainty like shot noise (probably less correlated) (Knobelspiesse et al., 2012). Therefore, we assume two scenarios: 1) angular correlation only existed in reflectance measurement, not in DoLP measurement, 2) both reflectance and DoLP have angular correlations with the same strength. Since the actual amount of correlation is not known, we designed our studies with the assumed correlation in the synthetic measurement, but with either no information or full information on the correlation angle in the retrieval cost functions. Therefore, four scenarios are discussed in this study

as summarized in Table 3.2 denoted by C1 to C4. We will discuss whether better retrieval results can be obtained if accurate correlation angles are considered in the retrieval cost function and whether we can estimate correlation from retrieval residuals in Sec. 4.



**Table 3.** Four scenarios of simulated uncertainties are considered in the synthetic data and retrievals. C1 and C2 indicate simulated errors with correlation angle of $\theta_c$ added to synthetic reflectance data. C1 assumes the correlation property is unknown and no correlation is considered in the cost function model, but C2 assumes the correlation as in the simulated errors is known. Similarly, for C3 and C4 which considered correlated uncertainty in both reflectance and DoLP.

| Scenario | Reflectance (Measurement) | DoLP (Measurement) | Reflectance (Retrieval) | DoLP (Retrieval) |
|----------|---------------------------|---------------------|-------------------------|------------------|
| C1 | $\theta_c$ | 0 | 0 | 0 |
| C2 | $\theta_c$ | 0 | $\theta_c$ | 0 |
| C3 | $\theta_c$ | $\theta_c$ | 0 | 0 |
| C4 | $\theta_c$ | $\theta_c$ | $\theta_c$ | $\theta_c$ |

## 3.2 Retrieval uncertainties impacted by uncertainty correlation

Using the Monte Carlo error propagation (MCEP) method discussed in Gao et al. (2022), we compared both real and theoretical retrieval uncertainties with different correlation angle and testing scenarios as summarized in Table for both synthetic AirHARP and HARP2 measurements. Fig. 5 demonstrates the basic approach in MCEP by comparing the AirHARP retrieval results with

5    $\theta_c = 60°$ for Scenarios C3 and C4, where the real retrieval errors are sampled based their retrieval uncertainties. The real uncertainties in both the root mean square error (RMSE) and the mean average error (MAE) are larger when uncertainty is correlated (comparing (b) and (a)). The theoretical uncertainties are similar because in both cases the correlation angles are assumed to be zero. After considering the same correlation angle in the retrieval cost function model as shown in Fig. 5 (c), both theoretical and real uncertainties are reduced. The real and theoretical uncertainties are similar to each other as shown

10   in Fig. 5 (a), but agreement degrades when correlation is considered (Fig. 5 b,c). There are more real retrievals errors in the negative side as shown in Fig. 5 (c). This may be related to the convergence for cases with small AOD values as discussed in Gao et al. (2022). Although real uncertainties are generally larger than theoretical uncertainties, the difference are mostly associated with cases where AODs are small and underestimated relative to the truth values.





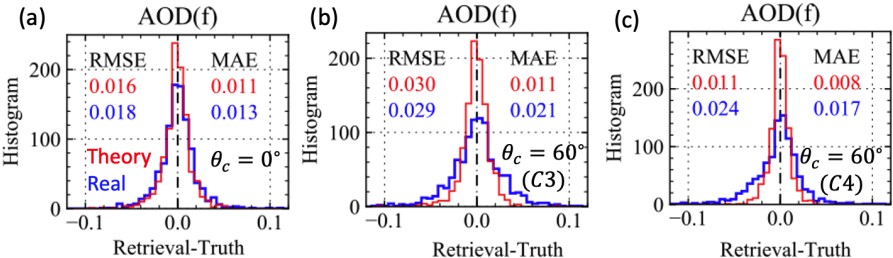

**Figure 5.** The histogram of the AirHARP retrieval errors for the fine mode AOD from theoretical and real uncertainty estimations based on the MCEP method. (a) for the case without any correlation in the uncertainty; (b) for the cases with correlation only in the synthetic data (C3), and (c). also in the retrieval cost function (C4) with correlation angle of $\theta_c = 60°$. Both mean average error (MAE) and root mean square error (RMSE) are computed for the theoretical and real errors indicated in red and blue text, respectively.

Both real errors and theoretical uncertainties have occasional outliers with large values due to poor convergence, and this has large impacts on the RMSE values. For example, in Fig. 5 (b) there are a few large estimated uncertainty values outside of the plot range that result in large RMSE for the theoretical error, almost comparable to the real uncertainty. Comparing MAE, however, shows that the theoretical value is much smaller than the real error, consistent with the histogram shape where the theoretical curve is narrower than the real curve. Give its robustness to outliers, MAE is used as the primary metric in this study.

Based on the MCEP method, we analyzed the retrieval uncertainties for synthetic AirHARP measurements from real errors and theoretical errors for various properties in Fig. 6, including the AOD, single scattering albedo (SSA), real part of refractive index ($m_r$), effective radius ($r_{eff}$), and effective variance ($v_{eff}$) for both fine and coarse modes and their combinations. $fvf$ is fine mode volume fraction. Ocean surface wind speed and Chlorophyll-a (Chla) concentration are also retrieved. As recommended by Seegers et al. (2018), the uncertainty of Chla is represented by a log-transformed metric:

$$\text{MAE(log)} = 10^Y \tag{15}$$

$$Y = \frac{1}{M}\sum_{i}^{M}[\log_{10}(\text{Chla}_{i,\text{retrieval}}) - \log_{10}(\text{Chla}_{i,\text{truth}})] \tag{16}$$

where M is the total number of samples which equals to the total number of synthetic measurement cases. The values of $\log_{10}(\text{Chla}_{i,\text{retrieval}}) - \log_{10}(\text{Chla}_{i,\text{truth}})$ are sampled for both the real and theoretical uncertainties similar to Fig.5 with detailed discussions provided in Gao et al. (2022). Furthermore, remote sensing reflectance at the four HARP wavelength are derived after conducting the atmospheric correction using the retrieved aerosol and ocean properties Gao et al. (2021a). Multiple set of random errors are sampled and averaged to estimate the impact of sample size in estimating retrieval uncertainties and shown in Fig. 6.

The real retrieval uncertainties for Scenario C3, in which correlation is considered in the simulated errors but not in the retrieval cost function, are found to be always increasing with the correlation angle. When the correct correlation angle is considered in the retrieval cost function (C4), the real retrieval uncertainty increases until $\theta_c$ reaching 10° to 20° and then



slightly decreases at higher correlation angles, for most retrieval parameters. The theoretical uncertainties for scenario C4 are similar to the real uncertainties for $\theta_c < 10°$ except for refractive index, for which the theoretical values are almost half that of the real uncertainties. When $\theta_c > 10°$, the theoretical uncertainties then decrease much faster than the real uncertainties as $\theta_c$ increases. The difference is mostly coming from the retrieval cases which underestimate the truth values as shown in Fig. 5.

5    Noted that the real retrieval uncertainties are mostly larger than the theoretical uncertainties without any correlation within the range of $\theta_c < 120°$. However theoretical uncertainties predict that the retrieval uncertainties increase slightly with $\theta_c$ around $10°$ (or $r = 0.9$), then decrease. Its values can be smaller than those with zero correlation when $\theta_c$ is larger than around $20°$ .

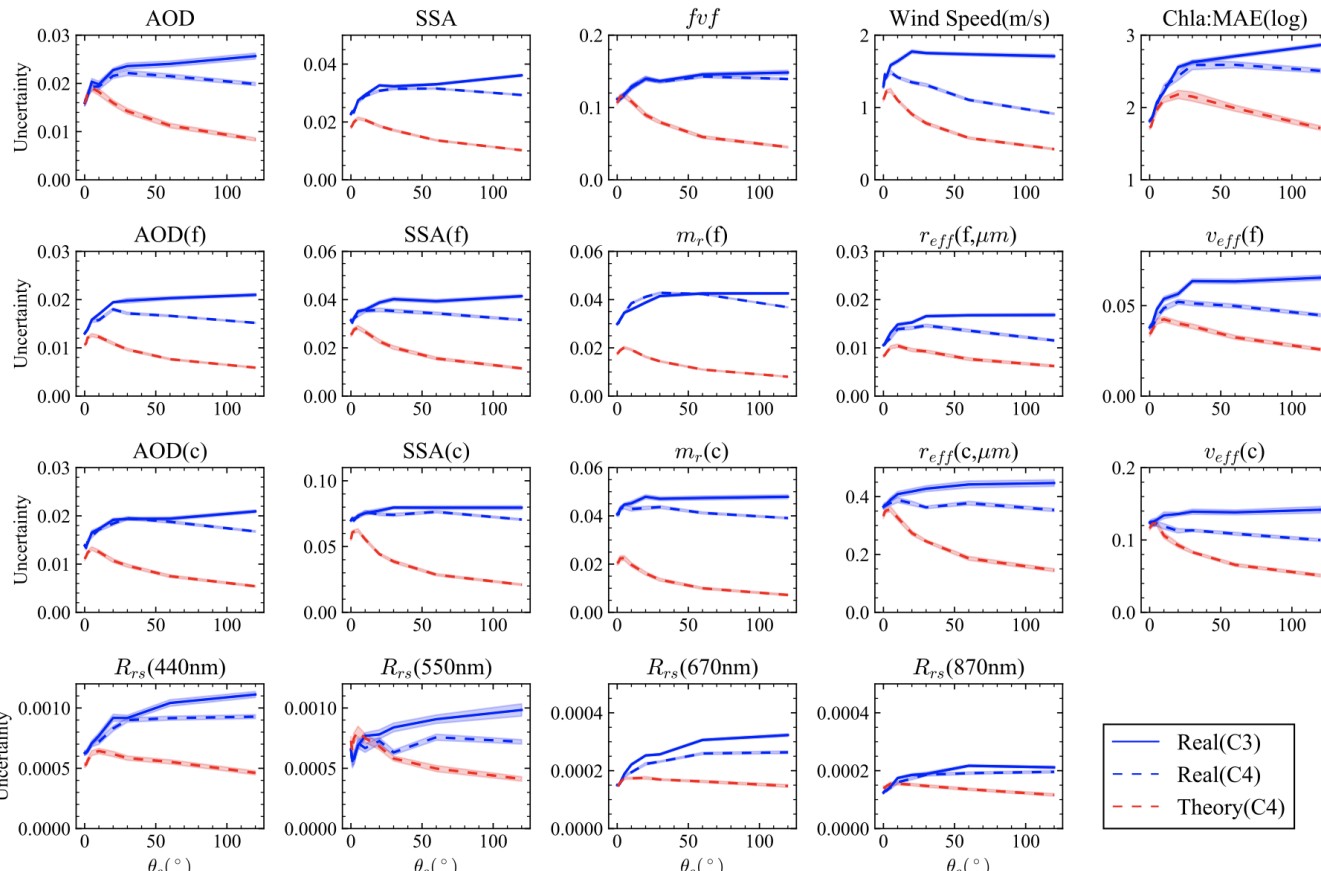

**Figure 6.** Retrieval uncertainties averaged for the AOD within [0.01,0.5] based on the MCEP method for synthetic AirHARP measurements. 10 sets of random samples are conducted, with the mean values shown in the plot and standard deviation indicated by the width of the shaded lines. The blue lines indicate the real uncertainty for Scenario C3 and Scenario C4 (Table 3.2), and the red line indicates the theoretical uncertainty for Scenario C4.

The results for HARP2 are similar to that of AirHARP as shown in Fig. 7. The overall HARP2 retrieval uncertainties are slightly smaller than AirHARP retrieval uncertainties, but the difference is mostly within 20% for the theoretical uncertainties,





and mostly within 10% for the real uncertainties as shown in Fig. 7 for both $\theta_c = 60°$ and $\theta_c = 60°$. Although HARP2 measures less viewing angles at 440, 550, and 870nm bands, its better DoLP accuracy still results in slightly smaller uncertainties in most cases. Note that the retrieval accuracies also depend on the total number of viewing angles used, and the range of scattering angles as discussed in (Gao et al., 2021b).

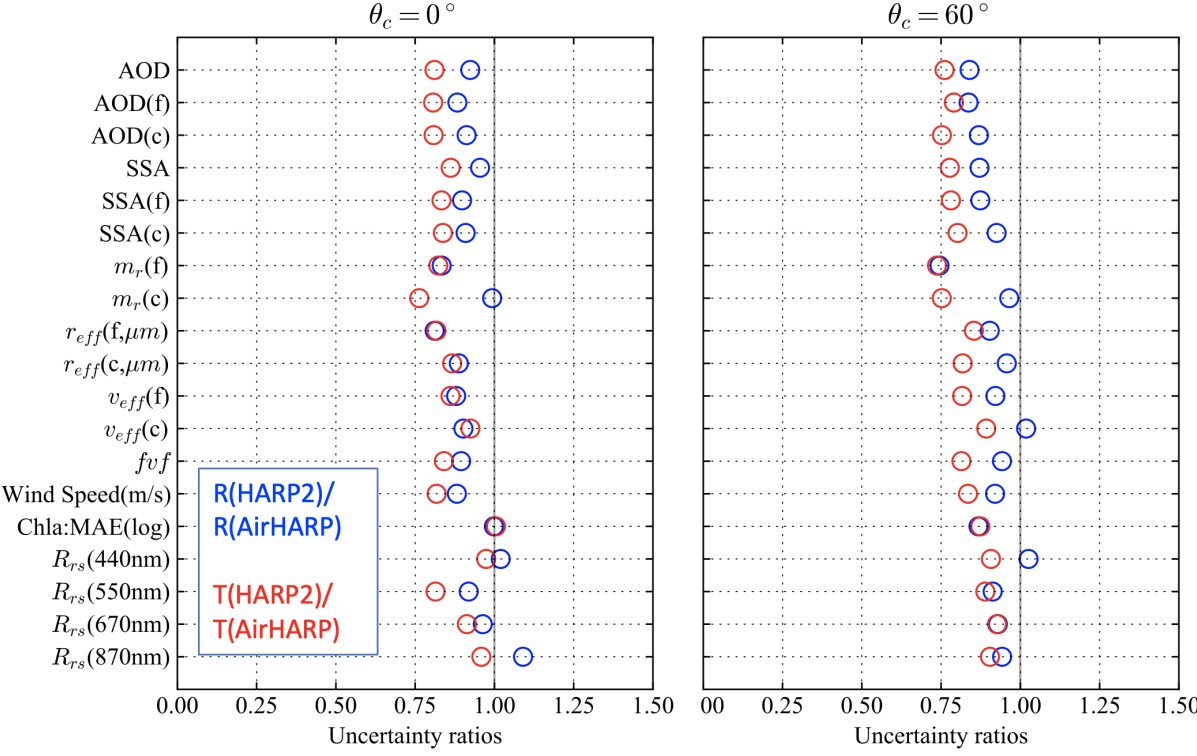

**Figure 7.** The ratio of the real (R) and theoretical (T) retrieval uncertainties between the AirHARP and HARP2 measurements for correlation angles equal to zero and 60°. In the legend, R(HARP2)/R(AirHARP) denotes the ratio of real retrieval uncertainties between HARP2 and AirHARP, and T(HARP2)/T(AirHARP) denotes the ratio of theoretical uncertainties.

5    To understand more quantitatively how the correlation angle impacts both AirHARP and HARP2 retrievals, the ratios of the real uncertainties between Scenarios C3 and C4 are represented as R(C4)/R(C3) for each retrieval quantity as shown in Fig. 8. Note that both C3 and C4 considered uncertainty correlations in both reflectance and DoLP, but only C4 considered the same correlation in its retrieval cost function and C3 assumes no correlation in its retrievals. Therefore, the ratio R(C4)/R(C3) represents how much retrieval uncertainties can be reduced if the correct amount of correlations are known in the retrieval

10    process. As shown in Fig. 8, the ratio is close to 1 for most parameters for $\theta_c = 10°$, with slightly larger impacts for effective variance for both fine and coarse mode as well as for wind speed uncertainty. Such ratios almost double for $\theta_c = 60°$ for both AirHARP and HARP2 measurements. When comparing the theoretical uncertainty T(C4) with R(C3), their ratio reduces to a value of 0.5 to 0.7 for $\theta_c = 10°$ for aerosol properties, and further decrease to 0.3 to 0.5 with $\theta_c = 60°$. The impacts





on the remote sensing reflectance are generally smaller for $\theta_c = 10°$, but show significant impacts (as small as 0.5) for $\theta_c = 60°$. These results demonstrate that there are potentially more rooms to reduce the retrieval uncertainty as predicted by the error propagation theory represented by Eq. (6), which worth future investigations. The largest gap between the real and theoretical uncertainties is still for refractive index in both fine and coarse mode same as observed in Gao et al. (2022). The gaps between the real and theoretical uncertainties increase with increasing $\theta_c$, which indicates degrading retrieval performance in the presence of correlated uncertainties.

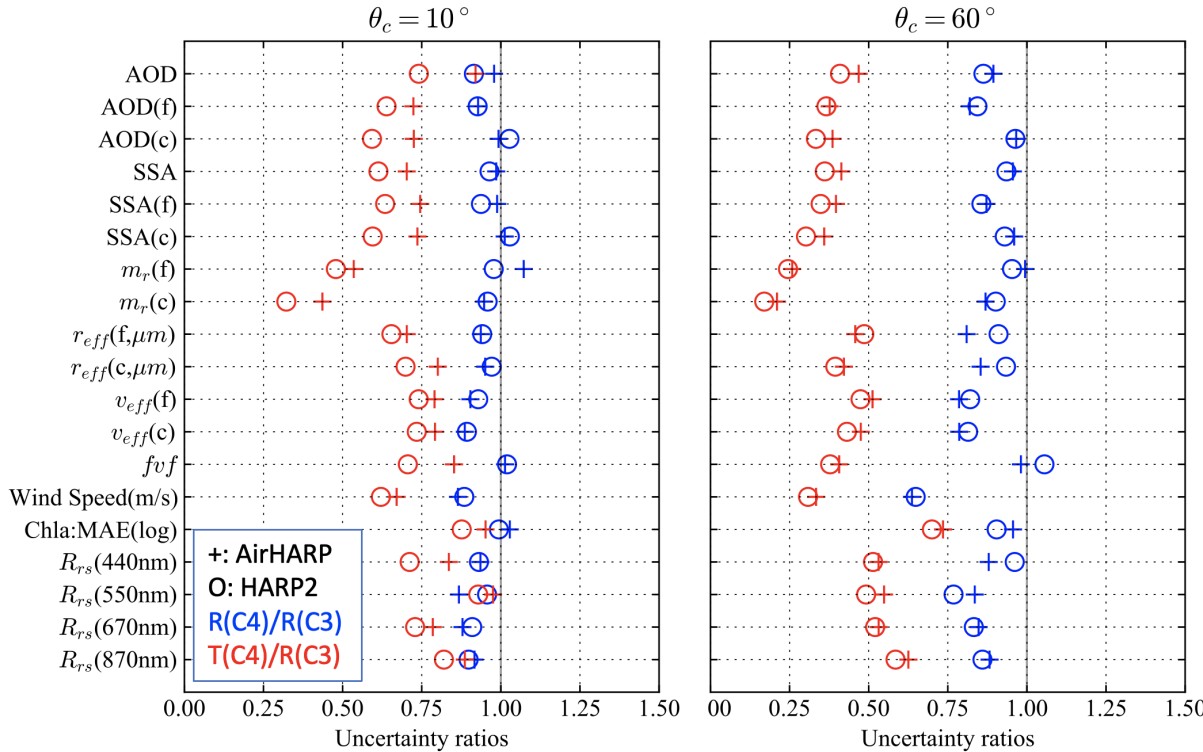

**Figure 8.** The ratio of real retrieval uncertainties between Senarios C4 and C3 as denoted in the lengend by R(C4)/R(C3) for AirHARP and HARP2 measurements with correlation angle equal to $10°$ and $60°$, similarly the real and theoretical uncertainties for C4 are compare with the real uncertainties for C3 denoted as T(C4)/R(C3).

To understand retrieval performance when uncertainty correlation is only in reflectance for Scenarios C1 and C2, similar ratios to Fig. 8 are plotted in Fig. 9. The ratios between the real uncertainties are almost always equal to a value of 1, with slightly larger impacts (less than 10%) for $\theta_c = 60°$. However, the theoretical errors are much smaller, with values ranging from approximately 0.75 for $\theta_c = 10°$ to 0.5 for $\theta_c = 60°$, and with even smaller values observed for refractive index. This suggest that correlation in reflectance alone has potential to be harvested to improve retrieval performance, but is even harder to be realized in real retrievals from current algorithms.



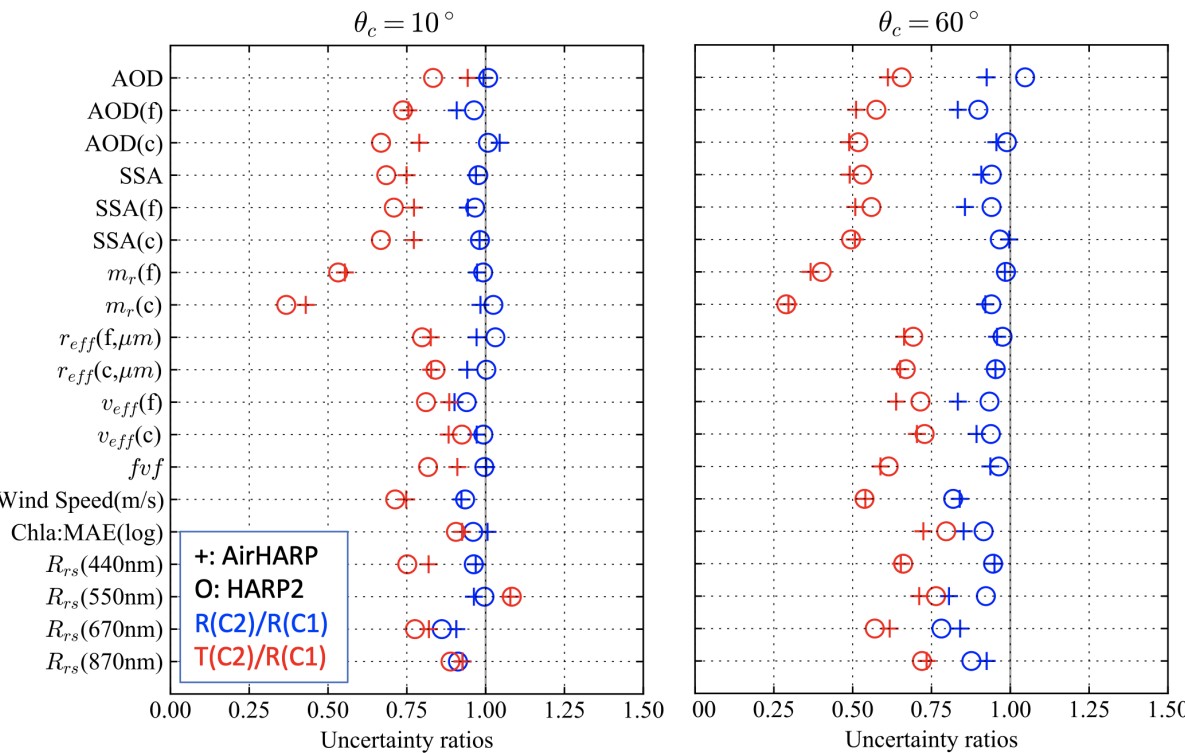

**Figure 9.** Similar to Fig. 8, but for the Scenarios of C1 and C2 when correlations are only in reflectance uncertainties.

## 4 Estimating correlation from residual analysis

### 4.1 Cost function and fitting performance

As discussed in Gao et al. (2021b), the retrieval cost function can be characterized well by the $\chi^2$ distribution for synthetic data, and for real data after removing anomalies such as cirrus cloud. Note that the cost function in Eq. (1) is called a $\chi^2$

5  function, but its histogram may not always follow a $\chi^2$ statistical distribution, which depend on how well the fitting residuals can represent the real uncertainty and the degree of freedom of the $\chi^2$ distribution (Gao et al., 2021a). Therefore, the histogram of successful retrievals is a useful indicator on how well the retrieval residuals compare with the assumed input uncertainty model (Rodgers, 2000). In this study, we found that the retrieval residuals can be represented well by the $\chi^2$ distribution with a degree of freedom equal to the total number of measurements used ($N$), which is twice of the total number of viewing angles

10  ($N_v$) when there is no correlation, as shown in Fig. 10. Note that the degree of freedom is defined for the $\chi^2$ distribution (Gao et al., 2021b).



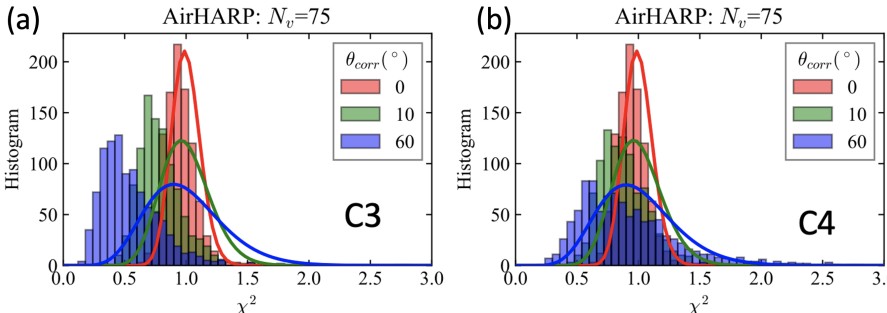

**Figure 10.** Cost function histogram with correlation angle of $0°$, $10°$ and $60°$ for Scenarios C3 (a) and C4 (b). The average number of viewing angles from the 1000 simulation cases, after removing sunglint, is $N_v = 75$ for the AirHARP measurement. The red line indicates the $\chi^2$ distribution with a degree of freedom of $2N_v = 150$. The green and blue lines indicate the $\chi^2$ distribution with a reduced degree of freedom of 40 and 20 fitted to the corresponding histogram.

However, the cost function histogram shifts to a smaller value for Scenario C3 (Fig. 10 (a)), where correlated uncertainty is included in the synthetic data but no correlation is considered in the retrieval cost function. One cause of the shift is due to overfitting of the data, which results in smaller residuals and larger real retrieval uncertainties as shown in Fig. 6. When the appropriate correlation is considered in the retrieval cost function as shown in Fig. 10 (b) for Scenario C4, the histogram

approaching to a $\chi^2$ distribution much closer, but with a smaller degree of freedom. For example, in the right panel in Fig. 10, a degree of freedom of 40 and 20 are found to better fit the cost function histogram with $\theta_c = 10°$ and $\theta_c = 60°$. This suggests that the correlation in the uncertainty reduces its degree of freedom.

To understand how much overfitting impacts retrievals under different strength of correlations, we compare the standard deviation of the retrieval residuals with the original simulated uncertainty, for all 1000 cases. For reflectance, we consider the

10 ratio between the simulated uncertainty and the reflectance as a convenient way to compare with the 3% (or 0.03) uncertainty model for reflectance (Fig. 11). For the simulated uncertainty, the value of 0.03 is confirmed, however, the retrieval residuals become smaller with increasing $\theta_c$ for C3. The 670 nm band shows the largest reduction, where a ratio of 0.025 is observed for $\theta_c = 10°$, reaching 0.015 for $\theta_c = 120°$, which may be due to the large number of angles (60) in the 670nm bands versus the other bands (10 or 20). This behavior indicates overfitting, where the uncertainties are partially removed as real signals.

When the correct correlation is considered in the model, the resulting is much like the assumption, with very slight indication of overfitting (within 0.005 in most cases). For the DoLP there are even larger indications of overfitting for the 670nm band as compared with other bands for C3, but reduces to around 0.01 for C4 for AirHARP. The results for HARP2 are very similar, but with the assumed uncertainty for DoLP of 0.005 (not shown).




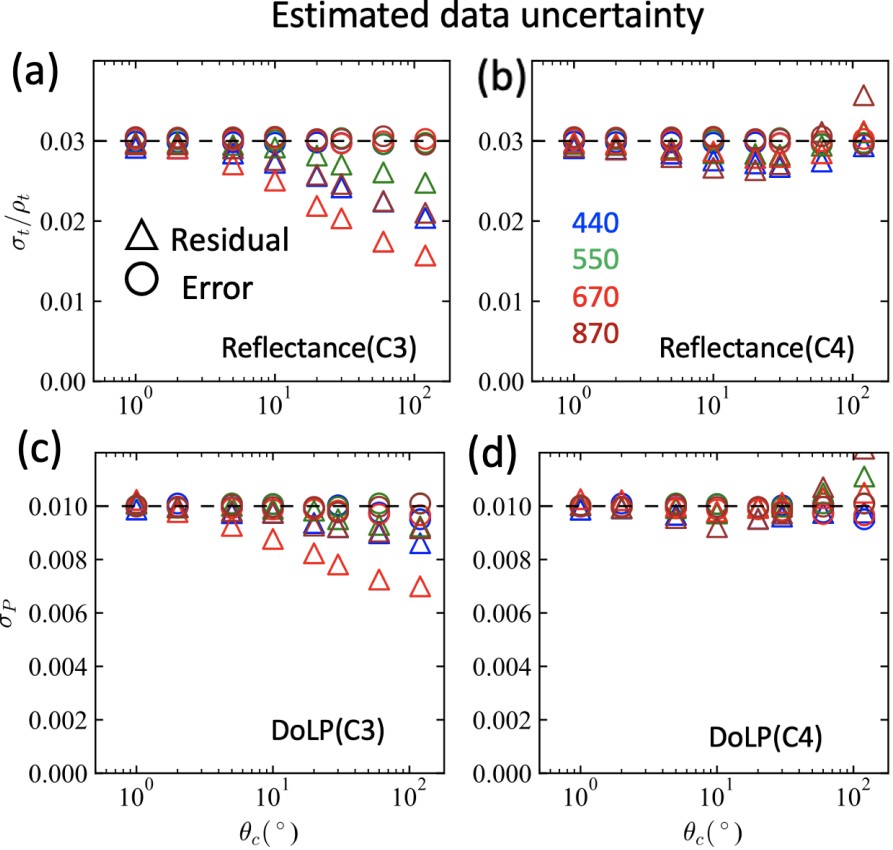

**Figure 11.** The standard deviation of simulated measurement errors and fitting residual for cases C3 and C4. The model uncertainty of 3% for reflectance and 0.01 for DoLP are indicated by dashed lines. Results on Cases C1 and C2 on reflectance is similar, but with estimated DoLP uncertainties closer to the 0.01 line.

## 4.2 Correlation estimation results with synthetic data

Different amount of over fitting also removed partially the correlation in the fitting residuals. As shown in Fig. 12 (a,b), the estimated correlation is smaller than the truth. The estimated $\theta_c$ for 670 bands are smaller than other bans which is consistent to previous study where this bands overfit the data the most. Green bands seems showed the best results which estimate $\theta_c < 10°$

5   $(r = 0.90)$ well, but underestimate true $\theta_c = 60°$ $(r = 0.983)$ as $20°$ $(r = 0.951)$ and $\theta_c = 120°$ $(r = 0.0992)$ as $30°$ $(r = 0.967)$. Note that the value of correlation parameter $r$ become asymptotically approaching to 1 and harder to be distinguished with a finite length of data. Meanwhile, we also compute the correlaton angles from the added simulated errors, which also resulted to a smaller correlation angle comparing to the truth but much closer than using the residual data. This may due to the finite length of the measurement which is $90°$ after the glint is removed (total $120°$). When the true correlation angle is

10   considered in the model (C4), the estimated correlation angle is improved. The results on DoLP is slightly better where the retrieval residuals seem estimate the results from simulated errors in a similar way for both C3 and C4 as shown in 12 (c,d).





In real data, the correlation strength may be different for reflectance and DoLP, it would be useful to estimate the correlation angles for all the four bands and both reflectance and DoLP, and analyze the difference by comparing with the synthetic data study.

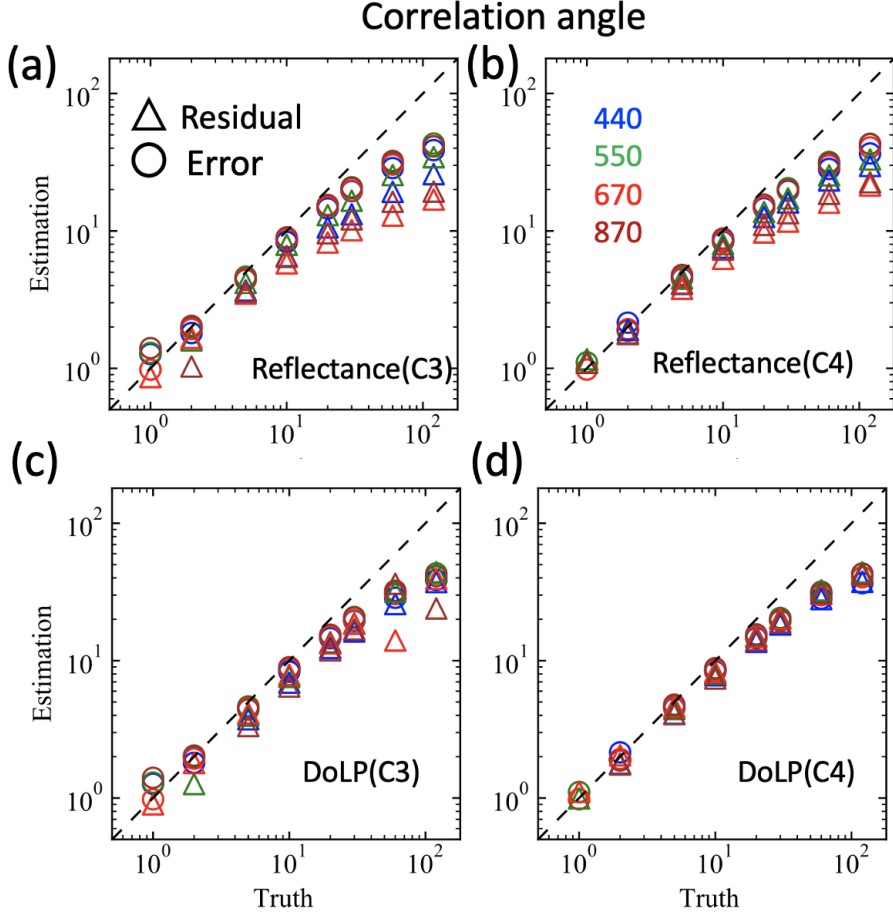

**Figure 12.** The estimated correlation angle $\theta_c$ for Scenarios C3 and C4 from the simulated measurement errors and fitting residuals for reflectance and DoLP are compared with the truth values. Dashed line indicates the 1:1 line. Results on Scenarios C1 and C2 on reflectance is similar, but with estimated correlation angle close to zero since no correlation is considered in DoLP (not shown).

### 4.3 Correlation estimation results with real AirHARP data

5    The Aerosol Characterization from Polarimeter and Lidar (ACEPOL) field campaign was conducted from October to November of 2017 with the NASA's ER-2 aircraft at a high altitude approximately 20 km (Knobelspiesse et al., 2020). Measurements over a variety of scenes are conducted from four MAPs: AirHARP, AirMSPI, SPEX airborne, and RSP; and two lidar sensors: HSRL-2 (Burton et al., 2015) and CPL (the Cloud Physics Lidar) (McGill et al., 2002). There is a total of five AirHARP ocean scenes available in the ACEPOL measurements with retrieval uncertainties studied by Gao et al. (2022) without considering



angular uncertainty correlation. Gao et al. (2021a) reported that the retrieval results on both aerosol and ocean color signals have are found to be in good agreement with the AERONET Ocean Color site (Zibordi et al., 2009). However, the cost function histogram was much wider than expected due to the impacts of cirrus clouds. After removing the cirrus cloud impacts from the multiple-angle measurement using an adaptive data screening method, the cost function histogram improved significantly with

much higher similarity with a $\chi^2$ distribution (Gao et al., 2021b). In this study, the same adaptive data screening methods are applied on all the five AirHARP ocean scenes, which removes cirrus clouds and other anomalies that could not be represented adequately by the current forward model. The resulting total number of measurements, including both reflectance and DoLP, are shown in Fig. 13, where the spatial distribution of pixels with many valid viewing angles is not uniform.

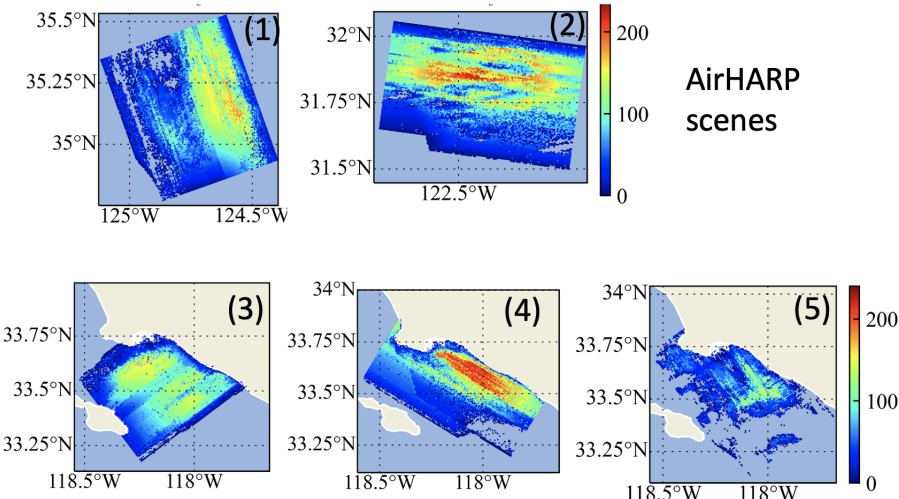

**Figure 13.** The number of total viewing angles (N) considering both reflectance and DoLP for the five AirHARP ocean scenes from ACEPOL.

Correlation properties from real measurements are difficult to quantity. However, we showed in Sec. 2.5 that as demonstrated

by last section, we showed autocorrelation analysis on the retrieval residual can be used as a good estimator when correlation is not too strong with consistent behaviors to derive the correlation parameters. Therefore, using the ACEPOL data, we can estimate angular uncertainty correlation from retrieval residuals. From each scene we selected 200 pixels with the most available number of angles, which are clustered together around the region with maximum number of measurements as shown in Fig. 14. The retrieval residuals data were then normalized as discussed in Sec. 2.5 to remove impacts by the non-uniform mean and

variance, which are often observed from the real data residuals. The autocorrelation and partial autocorrelation are calculated to access whether the AR(1) model is sufficient, with examples shown in Fig. 14, for the fitting residuals of reflectance and DoLP from the Scene 3 at 670nm band. Partial autocorrelation for reflectance showed similar results for from the synthetic data in Fig. 3 (b) with only the first order term prominent, which suggest that the AR(1) model is sufficient to describe the fitting residual for reflectance. Fig. 3 (d) for DoLP shows that higher order terms may also contribute to the uncertainty model; however, the overall correlation strength is small. From these plots, we can estimate the correlation parameters $r$ following Eq.



(14) where $\tilde{R}_1$ corresponds to the first order point in the autocorrelation plots. The values of $r$ are approximately 0.9 and 0.7 for reflectance and DoLP, respectively, corresponding to a correlation angles of approximately $10°$ and $15°$ following Eq. (5).

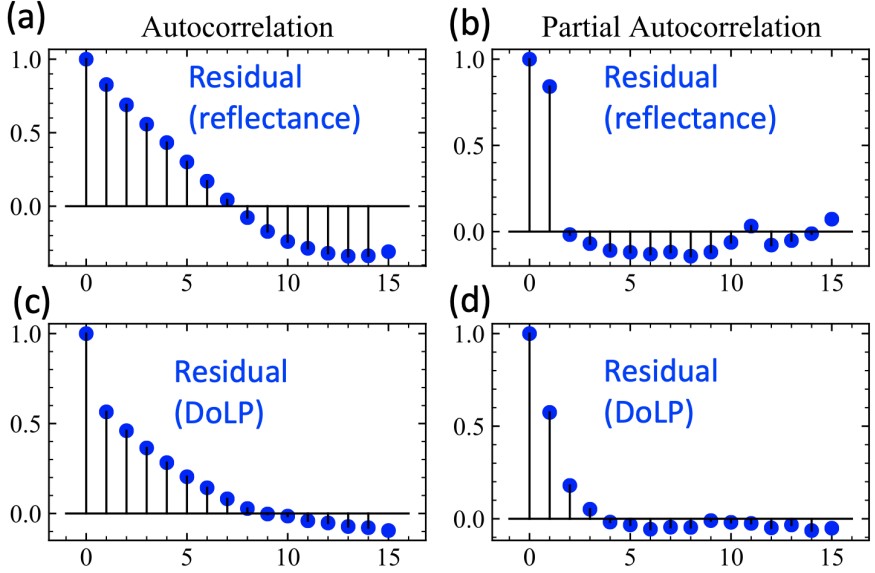

**Figure 14.** Autocorrelation (a and c) and partial autocorrelation (b and d) for the fitting residuals of reflectance and DoLP from Scene 3 at 670nm bands.

We analyzed the fitting residuals for all the five AirHARP scenes at the four bands with results summarized in Fig. 15. The minimal number of available angles used in the correlation estimation at each band are shown in Fig. 15(a). Note that different scenes have various number of angles removed due to the impacts of cirrus cloud or other anomalies. Those values are filled with zeros, which may reduce the strength of angular correlation. Therefore, the real correlation angle is likely larger than these values because only a subset of the total measurement data is used to estimate these angles. The estimated correlation angles for reflectance and DoLP varies mostly between $5°$ to $20°$. The correlation angles are smaller for band 670nm band, probably due to overfitting that partially removes the correlated errors as real retrieval signals, consistent with our observations based on synthetic data (Fig. 12). The correlation angles for the 440 and 550nm bands are largest, which may be mostly close to the truth. The correlation angles are generally larger for reflectance, with a value between $10°$ to $20°$ compared to DoLP with a value between $5°$ to $15°$, which suggests different amounts of correlation in the reflectance and DoLP data. Comparing the retrieval uncertainties for $\theta_c$ around $10°$ to $20°$ as shown in Figs. 6, the impact of the correlation to the real retrieval is small, but there is potential for large reduction of theoretical uncertainties by 25% to 50% if correct correlation is considered in the retrieval cost function. This requires, however, that the retrieval algorithm is capable of achieving its optimal performance as described by its theoretical uncertainties. Furthermore, the information on the correlation properties is useful to parameterize realistic measurement uncertainties into synthetic data. The correlation angles for reflectance and DoLP are likely larger than $10°$ and $5°$, respectively, which correspond to an estimated correlation parameter of 0.9 and 0.8.




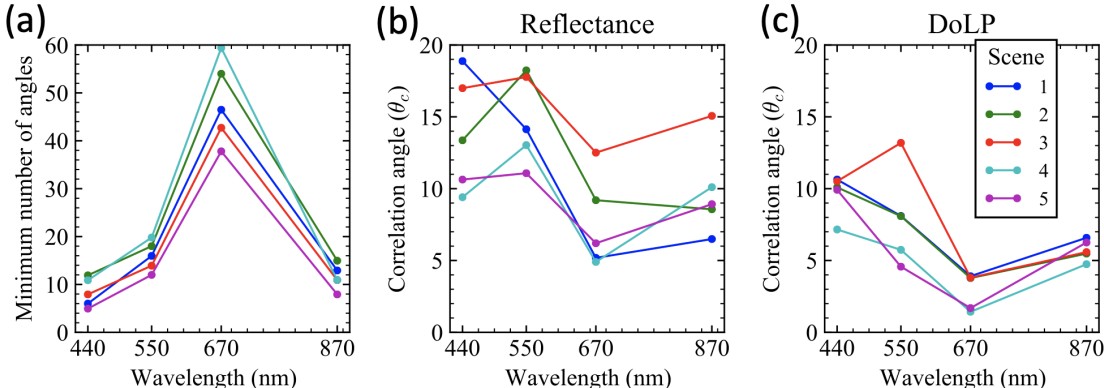

**Figure 15.** (a) The minimal number of angles used to estimate correlation parameters from each scene and wavelength; (b) and (c) the estimated correlation angles.

## 5 Discussions and conclusions

In this study, we evaluate the impacts of angular correlation on the retrieval uncertainties for various aerosol microphysical and optical properties, ocean surface properties, and water leaving signals. Theoretical uncertainties are derived based on error propagation and the real uncertainties are obtained through the comparison of retrieved and true values. The theoretical and real
uncertainties are compared and discussed. Only small angular correlation impacts are found on the real retrieval uncertainties unless the correlation strength is large (such as with correlation angle larger than $\theta_c > 10°$). The impacts vary with different retrieval parameters. Theoretical uncertainties are more impacted by angular correlation, which suggests the retrieval algorithm may not always converge to the global minima and that there is potential room for algorithm improvement.

Studies on the fitting residuals from both synthetic data and real AirHARP measurements were conducted. Autocorrelation
is useful to estimate the angular correlation, though it tends to underestimate when the correlation strength is strong and thus overfitting of the measurements is likely. Analysis on the real data showed that the angular correlation is stronger in the reflectance data than the DoLP, which makes sense because we expect that DoLP is less sensitive to systematic uncertainties that are more likely to be correlated. Partial autocorrelation analysis suggest that the uncertainty model considering a linear Markov process (AR(1)) is sufficient for reflectance, but may need to be further studied for DoLP. From AirHARP retrieval
residual analysis, the correlation angles for reflectance and DoLP are estimated to be larger than $10°$ and $5°$, corresponding to correlation parameters larger than 0.9 and 0.8, respectively

This work intends to provide basic methodology to analyze the measurement uncertainties with angular correlations, but the methods can also be applied in the spatial and spectral domains that may be more appropriate for other instruments. There are several remaining issues that need discussion in future works:

– Application to real data



It is complex to analyze real data as we discussed in previous sections. The major challenges and possible issues that may impact uncertainty correlation estimation are summarized below:

1. The retrieval is based on a forward model which also has uncertainties, a portion of which may be correlated. This uncertainty will contribute to the fitting residuals and may impact correlation analysis, but it is difficult to quantify.

2. The fitting residuals are often not stationary with uniform mean and variance. To reduce this issue, the residuals are normalized, but it would be valuable to analyze how the mean value and variance depend on the angle, as this may provide insight into the modeling uncertainties.

3. Some residuals are not continuous with angle due to removed cirrus clouds, which may reduce the correlation.

4. Synthetic data analysis has demonstrated that the retrieval is likely to overfit the data when the correlation is strong.

5. The angular grids for HARP measurements are slightly non-uniform, which is likely to further reduce the correlation strength from auto-correlation analysis. To evaluate impacts of this feature, an uncertainty model considering the impact of the real angular grids need to be built. But since the variation of the angular grids are less than $1°$ (670nm band) or $2°$ (other bands), which may impact more the cases with small correlation angles.

– Lab calibration

Although the correlation strength is estimated from fitting residuals from real AirHARP measurements, it may be only used for qualitative discussions due to various issues discussed above. To obtain the actual correlation properties, lab characterizations are desired to separate measurement characteristics. Lab measurement signals may also be evaluated through autocorrelation and partial correlation functions, or more general ARMA models. It is also interesting to discuss the possible impacts in the uncertainty model due to the binning and collocation that happen in later processing steps.

– Correlation strength as a fitting metric

Due the limitation discussed above, our analysis on the fitting residuals may only provide the lower boundary for the correlation strength in terms of correlation parameter ($r$) or correlation angle $\theta_c$. Since the correlation strength represents properties of the fitting residuals, it can be also used as a metric to represent retrieval fitting performance, together with the cost function $\chi^2$ and variance of the fitting residuals as discussed in Sec. 4.1.

– Signal correlation vs uncertainty correlation

Nature as measured by the instrument and expressed in the forward model has inherent correlation, which becomes part of the retrieval process. The phenomena we observe tend to be only slowly changing with respect to view angle, and thus measurements at different view angles do not necessarily express retrieved parameters independently. This correlation is related to the actual signal in the measurement rather than its uncertainties. This type of correlation is captured by the Jacobian matrix. The overall information content of the measurements with respect to the set of retrieval parameters are determined by both the Jacobian matrix and the correlated uncertainty model as further discussed in Appendix B.





– Future retrieval algorithm development

The pixel-wise theoretical uncertainties achieve a reasonably good performance to represent real retrievals when no correlation is presented. Their performances on various retrieved geophysical properties are quantified by comparing with the real retrieval errors. The difference grows bigger when the angular correlation is stronger, which suggests convergence to local minima and indicates that more development is needed to improve the retrieval optimization.

## Appendix A: Input parameters of the neural network forward model

A total of 15 parameters are used as input of the forward model as discussed in Sec. 2.1 as listed in Table A1. The solar zenith ($\theta_0$), viewing zenith angle ($\theta_v$), viewing azimuth angle relative to solar direction ($\phi_v$), and ozone density ($n_{O3}$) are assumed as known input. All the other 11 parameters are retrieval parameters in the FastMAPOL algorithm, including the aerosol volume density for each sub mode ($V_i$), the real ($m_r$) and imaginary parts ($m_i$) of the refractive index for fine and coarse mode aerosols, ocean surface wind speed ($w$) and Chloraphyll a concentration (Chla).

**Table A1.** Parameters used to train the FastMAPOL forward model. The minimum (min) and maximum (max) values of each parameter are also shown.

| Parameters | Unit | Min | Max |
|---|---|---|---|
| $\theta_0$ | ° | 0 | 70 |
| $\theta_v$ | ° | 0 | 60 |
| $\phi_v$ | ° | 0 | 180 |
| $n_{O3}$ | DU | 150 | 450 |
| $V_1$ | $\mu m^3 \mu m^{-2}$ | 0 | 0.11 |
| $V_2$ | $\mu m^3 \mu m^{-2}$ | 0 | 0.05 |
| $V_3$ | $\mu m^3 \mu m^{-2}$ | 0 | 0.05 |
| $V_4$ | $\mu m^3 \mu m^{-2}$ | 0 | 0.19 |
| $V_5$ | $\mu m^3 \mu m^{-2}$ | 0 | 0.58 |
| $m_{r,f}$ | (None) | 1.3 | 1.65 |
| $m_{r,c}$ | (None) | 1.3 | 1.65 |
| $m_{i,f}$ | (None) | 0 | 0.03 |
| $m_{i,c}$ | (None) | 0. | 0.03 |
| $w$ | $ms^{-1}$ | 0.5 | 10 |
| Chla | $mg \cdot m^{-3}$ | 0.01 | 10 |





## Appendix B: Eigenvector decomposition in error covariance matrix

Sec. 2.4 discussed that the error covariance matrix with non-diagonal terms can be diagonalized through eigen-decomposition as shown in Eq. (7). The original measurements can be transformed into a new space without correlations with uncertainty variance described by the eigenvalues in the diagonal matrix $\mathbf{D}_\epsilon$ denoted as $d_i^2$. To understand how the uncertainties vary with different correlation strength, the square root of the diagonal term in $\mathbf{D}_\epsilon$ are used to represent the new measurement uncertainties. Results for different correlation angles are shown in Fig. B1.

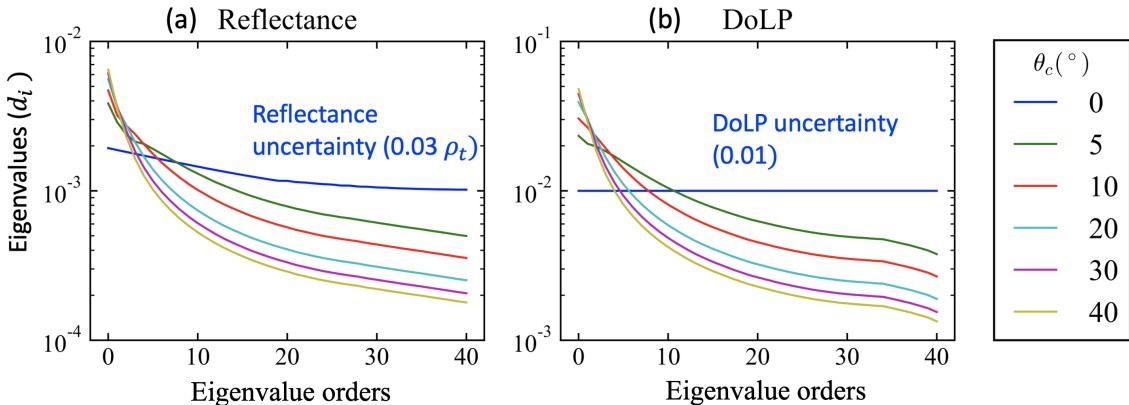

**Figure B1.** The uncertainties for reflectance and DoLP of the error covariance matrix at 670nm band after eigenvector decomposition as discussed in Sec. 2.4. These values correspond to the square root of the diagonal values in matrix $\mathbf{D}_\epsilon$ from Eq. (7).

The uncertainties in the new measurement space ($\mathbf{y}'$) show components both larger and smaller than the original uncertainties from different combinations of measurements. The corresponding Shannon information content (SIC) (Rodgers, 2000) are defined as

$$SIC = \frac{1}{2}\ln|\mathbf{S}^{-1}\mathbf{S_a}| \tag{B1}$$

$$= \frac{1}{2}\ln|\mathbf{S_a^{1/2}}\mathbf{K'^T}\mathbf{D}_\epsilon^{-1}\mathbf{K'}\mathbf{S_a^{1/2}}+\mathbf{I}| \tag{B2}$$

where the error covariance matrix $\mathbf{S}$ and the a priori matrix $\mathbf{S_a}$ in Fig. B2 are from Eq. (6). Jacobian matrix and the error covariance matrix are converted into the diagonal space as shown in Sec. 2.4, and are used to represent the SIC in Eq. (B1). Different correlation strength will lead to different unitary matrix which is used to transform the Jacobian and error covariance matrix (Eq. (7)). We analyzed the SIC and retrieval uncertainties for a sequence of correlation angles from 0 to 120 degrees. As shown in Fig. B2 , when the correlation strength is strong the SIC is increasing with theoretical retrieval uncertainties estimated from error propagation such as for fine mode refractive index, wind speed and Chla decreasing. However, when the correlation is relatively weak, SIC decreases and uncertainties increase with the correlations. The correlation angles with maximum uncertainties are different with different retrieval parameters, but generally fall within 30 degrees. This behavior may relate how the measurement uncertainties are mapped to the retrieval parameters space through Jacobian matrix. Similar behavior of the



SIC has been reported by (Knobelspiesse et al., 2012) for the RSP measurement. Understanding of the uncertainty properties such as correlation strength are useful to further exploit the information in the measurements.

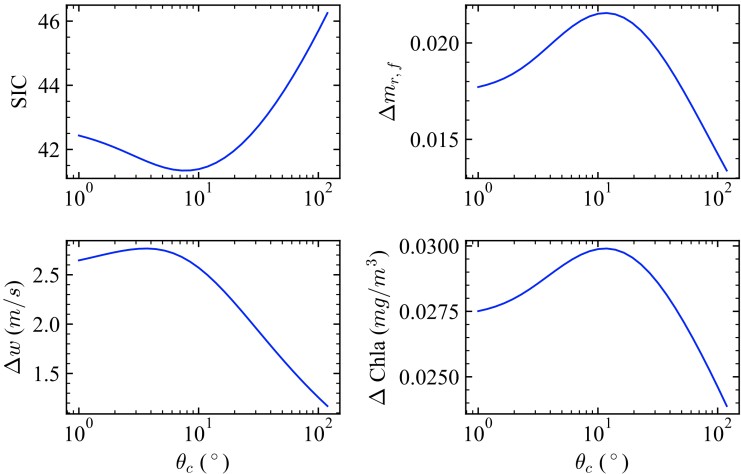

**Figure B2.** The Shannon information content (SIC) and corresponding theoretical retrieval uncertainties for refractive index ($m_{r,f}$), wind speed ($w$) and Chla with respect to various correlation angles. The results are for one example case with Chla $= 0.1 mg/m^3$, $w = 3.0 m/s$, $m_{r,f} = 1.55$ and AOD (550nm) $= 0.2$ same as the case discussed in (Gao et al., 2022) ( ID=201).

*Competing interests.* The authors declare no conflict of interest.

*Acknowledgements.* The authors would like to thank the ACEPOL teams for conducting the field campaign. The numerical studies are

5   conducted on the Poseidon supercomputer cluster at NASA Ocean Biology Processing Group (OBPG). We thank the OBPG system team for supporting the high-performance computing. We thank Zhonghuan Chen, Can Li, Andy Sayer, Amir Ibrahim, Jason Xuan, Yunwei Cui for constructive discussions.

   Meng Gao, Kirk Knobelspiesse, Bryan A. Franz, and Brian Cairns are supported by the NASA PACE project. Peng-Wang Zhai is supported by NASA (grant no. 80NSSC20M0227). The ACEPOL campaign has been supported by the NASA Radiation Sciences Program, with

10   funding from NASA (ACE and CALIPSO missions) and SRON. Part of this work has been funded by the NWO/NSO project ACEPOL (project no. ALWGO/16-09).

*Data availability.* The AirHARP data used in this study are available from the ACEPOL data portal (https://doi.org/10.5067/SUBORBITAL/ACEPOL2017/DATA001).





*Author contributions.* MG, KK, BF, P-WZ, BC formulated the study concept. MG generate the scientific data and wrote the original manuscript. P-WZ developed the radiative transfer code used to train the NN models. XX and VM provided and advised on the HARP data. All authors provided critical feedback and edited the manuscript.





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
