# Peer review of "The impact and estimation of uncertainty correlation for multi-angle polarimetric remote sensing of aerosols and ocean color"

_EGUsphere, 2022_

## Author Comment (AC1)

We appreciate the effort and time the reviewer has invested in reviewing our manuscript. We are grateful for the constructive feedback, which have improved the quality of our research. Please find our response below with revision details.

The topic of the paper is very important and interesting. The paper is, in general, well-written and quite easy to follow. I only have a few comments, including some technical ones. PXLX below indicates page X and line X.

Thanks for the positive comments.

P5L11 The atmosphere and ocean system is assumed to be a four-layer system. Is there any coupling effect between the layers being taken into account?

Yes, the radiative transfer simulation (RTSOS) fully considers the coupling between these four layers. We added the following information:

Original:

> "The atmosphere and ocean system are assumed to be a four-layer system."

Revised:

> "The atmosphere and ocean system are assumed to be a four-layer system and **the radiative transfer interactions among them are fully considered in the RTSOS model.**"

P5L15 The layer extends from the ocean surface to a height of 2 km. Which profile shape is used?

The aerosol layer is assumed to be uniform, and the molecular profile follows the US standard atmospheric profile. We revised our manuscript as follows:

Original:

> "…The third layer is an aerosol layer mixed with Rayleigh scattering. The layer extends from ocean surface to a height of 2km. The last layer contains atmospheric molecules from 2 km to the top of the atmosphere."

Revised:
> "…The third layer is an aerosol layer mixed with Rayleigh scattering. This layer extends from ocean surface to a height of 2km **with a uniform aerosol vertical distribution.** The last layer contains atmospheric molecules from 2 km to the top of the atmosphere. **The US standard atmospheric constituent profile is used to describe the molecular distributions (Anderson et al., 1986).**"

P5L27, 'An accuracy of less than 1% for reflectance and less than 0.003 for DoLP has been achieved' What is the uncertainty for all instrument-related issues for AirHARP and HARP2?

The instrument uncertainty in reflectance is 3% for both AirHARP and HARP2. For DOLP, the uncertainty is 0.01 for AirHARP and 0.005 for HARP2. These information is provided in a latter location in Section 3.1 :

> "Correlated errors for both the AirHARP and HARP2 instruments are generated according to the same 3% uncertainty for reflectance, but 0.01 in DoLP for AirHARP and 0.005 in DoLP for HARP2."

We made the following revision to introduce these uncertainties early:

Original:

> "The accuracy of the NN forward model is examined with an independent synthetic measurement dataset not used in training. An accuracy of less than 1% for reflectance and less than 0.003 for DoLP has been achieved (Gao et al., 2021a).
> "

Revised:

> "The accuracy of the NN forward model is examined with an independent synthetic measurement dataset not used in training. An accuracy of better than 1% for reflectance and better than 0.003 for DoLP has been achieved (Gao et al., 2021a).**The uncertainties of the NN forward model are less than the instrument uncertainties of AirHARP and HARP2 (~3% in reflectance, 0.01 in DoLP for AirHARP, and 0.005 in DoLP for HARP2).**
> "

P6L2 what is t in $\rho_t$ and $P_t$

Here we use 't' to indicate "total" signals, as this signal involves contributions from aerosol scattering, Rayleigh scattering, ocean surface contribution, etc. We revised the sentence as follows:

Original:
> "Measurement vector m includes both reflectance ($\rho_t$) and DoLP ($P_t$) with the total number of measurements of N, which has been used in previous studies (Gao et al., 2021a)."

Revised:
> "Measurement vector m includes both reflectance ($\rho_t$) and DoLP ($P_t$) **where the subscript t indicates the total signal measured by the instrument. The total number**

**of measurements, N, at each pixel includes contributions from both reflectance and DoLP, which has been used in previous studies (Gao et al., 2021a, 2021b)."**

P6L22, AR(1) works well for most cases. Under what conditions will AR(1) have potential problems?

Thanks for the question. We have revised the sentence to make it more clear. Note that AR model is only briefly mentioned here and more detailed analysis will be followed:

Original:

"However from our analysis of AirHARP measurements in Sec. 4.3, AR(1) works well for most cases."

Revised:

"However from our analysis based on the retrieval results from real AirHARP measurements, **AR(1) works well for most cases. Detailed analysis can be found in Sec 2.5 for theoretical basis, Sec 4.3 for real data applications, and Sec 5 for general discussions.**"

Then for Sec 2.5, we made the following edits:

Original:

"

However, the mean values and variance in the fitting residuals often vary with respect to the angular grids. This type of signal is classified as non-stationary and difficult to model (Priestley, 1983). To overcome this issue, the original residual data y is processed by removing its mean and normalizing by its standard deviation.

"

Revised:

"

However, the mean values and variance in the fitting residuals often vary with respect to the angular grids. This type of signal is classified as non-stationary and difficult **to study by the AR models** (Priestley, 1983). To overcome this issue, the original residual data y is processed by removing its mean and normalizing by its standard deviation.

"

Furthermore, we have detailed discussion in Section 5 with several potential challenges in using AR model and other correlation analysis tools (no revision here):

> "…
>     1. The retrieval is based on a forward model which also has uncertainties, a portion of which may be correlated. This uncertainty will contribute to the fitting residuals and may impact correlation analysis, but it is difficult to quantify.
>     2. The fitting residuals are often not stationary with uniform mean and variance. To reduce this issue, the residuals are normalized, but it would be valuable to analyze how the mean value and variance depend on the angle, as this may provide insight into the modeling uncertainties.
>     3. Some residuals are not continuous with angle due to removed cirrus clouds, which may reduce the correlation.
>     4. Synthetic data analysis has demonstrated that the retrieval is likely to overfit the data when the correlation is strong.
>     5. The angular grids for HARP measurements are slightly non-uniform, which is likely to further reduce the correlation strength from auto-correlation analysis. To evaluate impacts of this feature, an uncertainty model considering the impact of the real angular grids need to be built. But since the variation of the angular grids are less than 1 (670nm band) or 2 (other bands), which may impact more the cases with small correlation angles.
>     "

P7L15, how uncertain is such an assumption ,the retrieval parameters successfully converged to the global minima ' and what is the potential impacts on the error propagation and also on the retrieval results?

Since our retrieval algorithm involves many retrieval parameters, it is always a challenge to ensure that all parameters converge to the global minima. If the retrievals converge to a local minimum, the retrieval parameter can be less accurate, and the derivatives around the local minima may not represent the actual values near the global minima. We made the following edits:

Original:

> "The retrieval uncertainties estimated by error propagation (hereafter called theoretical retrieval uncertainty) as shown in Eq. (6) represent the optimal scenarios, with limitations such as the assumption that the retrieval parameters successfully converged to the global minima (more discussions in Sayer et al. (2020); Gao et al. (2022))."

Addition:

"...**Both the retrieval results and associated Jacobians can be less representative to the truth values and therefore lead to inaccurate error propagation and uncertainty estimation**."

P9L4-5, what is the value of a typical correlation angle for AirHARP and HARP2 near 660 nm?

Thank for the questions. Since the correlation properties for HARP instrument have not been characterized before, there is less quantitative information on a typical correlation angle value. However, in our analysis based on the retrieval analysis, the correlation angle for reflectance is likely larger than 10-20°. Since, there are also possible chances of underestimation as discussed in Sec 4.2, we choose a wider range of correlation angle in our study from 0 to 120°. In this section, we use 10° and 60° as representative value of correlation angle to demonstrate the properties of correlated errors.

Furthermore, Knobelspiesse et al (2012) has assumed a correlation parameter of 0.9 (corresponding a correlation angle of 10°) in the study of the angular correlation of RSP instrument. This is another reason, we select 10° as an example.

We have revised as follows:

Original:

"The correlated error samples with correlation angle of $\theta c = 10°$ (r = 0.9) and correlation angle of $\theta c = 60°$ (r = 0.98) are shown in Fig. 2 (a) and (c)."

Revised:

"**To demonstrate how angular correlations impact the errors,** the correlated error samples with correlation angle of $\theta c = 10°$ (r = 0.90) and correlation angle of $\theta c = 60°$ (r = 0.98) are shown in Fig. 2 (a) and (c). **A value of r=0.9 has been assumed in the study of RSP angular correlation by Knobelspiesse et al (2012).**"

On a related subject, there are several works study the correlation in different domain (retrieval parameters, spectral, spatial). We have summarized related work in the introduction and adding more reference below:

Original:

"Retrieval algorithms that exploit correlation information in retrieval parameters and measurement uncertainties have shown benefits in improving remote sensing capabilities. The Generalized Retrieval of Aerosol and Surface Properties (GRASP) algorithm retrieves multiple pixels simultaneously, while considering the spatial correlation of the retrieval parameters (Dubovik et al., 2014, 2021). Xu et al. (2019) developed a correlated multi-pixel inversion approach (CIMAP), which further considers the correlation between different retrieval parameters. Theys et al. (2021) developed a Covariance-Based Retrieval Algorithm (COBRA) based on an error covariance matrix estimated from

measurements with spectral correlation, applied their approach to sulfur dioxide (SO2) retrievals from the TROPOspheric Monitoring Instrument (TROPOMI) data, and demonstrated improved retrieval performance. "

Addition:

> **… To accurately evaluate pixel-level uncertainty in ocean color retrievals, spectral correlation associated with the uncertainty in top-of-atmosphere reflectance are also accounted for OLCI (Lamquin et al, 2013) and MODIS (Zhang et al, 2022) in the uncertainty propagation."**

Added reference:
- **N. Lamquin, A. Mangin, C. Mazeran, B. Bourg, V. Bruniquel, and O. F. D'Andon, "OLCI L2 Pixel-by-Pixel Uncertainty Propagation in OLCI Clean Water Branch," (ESA, 2013), p. 51.**

- **Zhang, M., A. Ibrahim, B. A. Franz, Z. Ahmad, and A. M. Sayer. 2022. "Estimating pixel-level uncertainty in ocean color retrievals from MODIS." Optics Express, 30 (17): 31415 [10.1364/oe.460735]**

P9L6, Errors start to form a longer range of correlation with smoother variations. Is the smoother pattern caused by the relatively small magnitude of the correlation angle of 60 degrees, or is it really true that an increase in the correlation angle leads to a decrease in the magnitude and pattern of the errors with respect to the viewing angles? Why did the author limit the viewing angles to 25, rather than 60?

Thank you for the question. A larger correlation angle indicates a longer range of correlation and appears with smoother variation also shown in Fig 2. However, the increase of correlation does not necessarily mean a decrease of the magnitude. As shown in Fig 2(b), the errors with stronger correlation will start to move together, but the overall magnitude can vary in a wide range.

We added more discussion here:

Original:
> "With larger $\theta$c the errors start to form longer range of correlation with smoother variations."

Revised:
> "With larger $\theta$c the errors start to form longer range of correlation with smoother variations. **Note that the overall magnitude of the errors can vary within the full range as described by the calibration uncertainties**"

Regarding the viewing angle at the right side of the plot in Fig 2, since there is strong glint in the right side, which cause issues in training the neural network models, we have removed the sunglint as shown in Fig 4, which corresponds to the partial removal of the angles in the right side.

The following sentence is added to the Fig 2 caption:

> ".. The right side of viewing angle ends around 25º due to the removal of sunglint as shown in Fig 4."

P11L6, Chla to Chl-a

Corrected into Chl-a. And checked the whole document.

P11L8 The range of [0.01, 0.5] for AOD sounds reasonable. However, as there are many plumes along the coastal regions, will such a restriction of 0.5 lead to a too-small error estimation for real measurements?

Thanks for the question. The range of [0.01, 0.5] is the nominal range of aerosol loading for ocean color remote sensing. The development of neural networks for cases with larger AOD will be a subject of future work.

Original:

> "The same sampling approach discussed in Gao et al. (2022) is conducted assuming that the aerosol optical depth (AOD) and fine mode volume fraction are uniformly distributed within [0.01, 0.5] and [0,1], respectively."

Addition:

> "… A larger range of AOD values will be needed for applying this algorithm to cases of smoke and plume events."

P13L5-7 The real uncertainties in both the root mean square error (RMSE) and the mean average error (MAE) are larger when uncertainty is correlated (comparing (b) and (a)) , it seems for (b), the real one is smaller (0.029 vs 0.03) as compared to the theoretical one?

Thank you for noticing the difference. Since the analysis is based on Monte Carlo sampling, there could be statistical fluctuations. To reduce the impact of such fluctuation (and corresponding uncertainties), in our latter discussion, we sampled 10 times, and take their average. As shown in Fig 6, the average real uncertainties are still mostly larger than theoretical uncertainties. Please find more detailed discussion for this technique in Gao et al (2022) (https://doi.org/10.5194/amt-15-4859-2022) . We have revised our discussion as follows to be more precise:

Original:

> "The real uncertainties in both the root mean square error (RMSE) and the mean average error (MAE) are larger when uncertainty is correlated (comparing (b) and (a))."

Revised:

> "The real uncertainties in both the root mean square error (RMSE) and the mean average error (MAE) are **mostly** larger when uncertainty is correlated (comparing (b) and (a)) **with exceptions possibly due to statistical fluctuations in the Monte Carlo sampling.**"

P14 L1, Both real errors and theoretical uncertainties have occasional outliers with large values due to poor convergence. Is there any link due to the assumption that the retrieval parameters successfully converged to the global minimum? (P7L15)

Yes, we also agree with the reviewer. The outliers could be related to the poor convergence not reaching the global minimum. We revised as follows:

Original:

> "Both real errors and theoretical uncertainties have occasional outliers with large values due to poor convergence, and this has large impacts on the RMSE values."

Revised:

> "Both real errors and theoretical uncertainties have occasional outliers with large values possibly due to convergence **to local minima instead of global minima,** and this has large impacts on the RMSE values."

P16 Title of Fig7, change zero to ‚0'

Corrected.

P16L4  change (Gao et al., 2021b) to Gao et al (2021b)

Corrected.

P17 L1, but show significant impacts (as small as 0.5) for θc = 60∘ . Is this due to the small value of theoretical uncertainty (red in Fig.6), which leads to small ratio? If so, I would suggest the author put more effort to explain Fig. 6.

Yes, we agree with the reviewer. The theoretical uncertainty decreases faster with stronger correlation. We added more discussion here:

Original:

"The impacts on the remote sensing reflectance are generally smaller for $\theta c = 10$ °, but show significant impacts (as small as 0.5) for $\theta^c = 60°$."

Revised:

"The impacts on the remote sensing reflectance are generally smaller for $\theta c = 10$ °, but show significant impacts (as small as 0.5) for $\theta^c = 60°$. **This is because the theoretical uncertainty with correlation angle decreases faster than the real uncertainties (see Fig. 6).**"

P19L3 overfitting of the data, have you check this issue with , test set '

Yes, this study is based on the synthetic simulation data with controlled error added to the simulation. To understand this overfitting effect we also computed the standard deviation of the retrieval residuals with results shown in Fig 11, which also shows smaller values than expected uncertainty model.

Original:

"One cause of the shift is due to overfitting of the data, which results in smaller residuals and larger real retrieval uncertainties as shown in Fig. 6."

Revised:

**"Smaller cost function values indicate smaller retrieval residual, which may be caused by overfitting of the data, and also possibly lead to the larger real retrieval uncertainties as shown in Fig. 6."**

More discussions are provided in the next section.

P19 L6, a degree of freedom of 40 and 20 are found to better fit the cost function histogram with $\theta c = 10°$ and $\theta c = 60°$, as compared to $\theta c = 0°$? Or the comparison between (a) and (b) in Fig. 10?

Yes, this refers to the Fig 10 (a) and (b). Here we are trying to find an approximated degree of freedom which can fit the cost function histogram when correlations are presented. We find that we have to reduce the original degree of freedom of 150 (150 measurements used) to a value of 40 and 20 for correlation angle of 10 and 60 degrees.

Original:

For example, in the right panel in Fig. 10, a degree of freedom of 40 and 20 are found to better fit the cost function histogram with θc = 10◦  and θc = 60◦,

Revised:

For example, in the right panel in Fig. 10, **chi^2 distribution with a degree of freedom of 40 and 20 are found to better fit the cost function histogram with θc = 10◦  and θc = 60◦, comparing with results using all the measurement degree of freedom (150).**

More information is also provided in the Fig 10 caption:

"The red line indicates the chi2 distribution with a degree of freedom of $2Nv = 150$. The green and blue lines indicate the chi2 distribution with a reduced degree of freedom of 40 and 20 fitted to the corresponding histogram."

P19L14, This behaviour indicates overfitting, where the uncertainties are partially removed as real signals. It is removed or considered?

Thank you for the suggestion. Here the errors is treated as real signal, and removed from the retrieval residuals by the model. We believe you can also say they are "considered" in the model. We revised the paragraph:

Original:

"This behavior indicates overfitting, where the uncertainties are partially removed as real signals."

Revised:

"This behavior indicates overfitting, where the errors are partially removed as real signals **and lead to reduced residuals**."

P22L17 Partial autocorrelation for reflectance showed similar results for from the synthetic data in Fig. 3 (b) with only the first order term prominent, which suggest that the AR(1) model is sufficient to describe the fitting residual for reflectance. However, we can see clear differences in the dependence on angular step k. Why is this?

The difference is likely due to the small higher correlation terms. The AR(1) model capture the major correlation behavior in the data. The higher order contributions are mostly small as suggested by the partial autocorrelation function.

Original:

"Partial autocorrelation for reflectance showed similar results for from the synthetic data in Fig. 3 (b) with only the first order term prominent, which suggest that the AR(1) model is sufficient to describe the fitting residual for reflectance."

Revised:

"Partial autocorrelation for reflectance showed similar results for the synthetic data in Fig. 3 (b) with only the first order term prominent, which suggest that the AR(1) model is sufficient to describe the fitting residual for reflectance, **with higher order contributions negligible**."

After reading the whole manuscript, I am thinking maybe the author should make a more detailed summary at P5L3-8 because there are many citations of their previous work, which requires quite some effort to check those very relevant publications. But I leave this comment open to the authors.

Thank you for the suggestion. The paragraph in page P5L3-8 is aimed to provide a high-level summary to our previous work, which focusing on the development of the retrieval algorithm and application of neural networks to improve speed and accuracy. The study in the error correlation is tested using the results from the retrieval algorithm, but not limited to a specific algorithm, and should be general in treating any other error correlations. Since we relied on the uncertainty quantification method developed in Gao et al. (2022), we provided more details and examples in Sec. 3.2. More details on the multi-angle cloud masking used in this study to reduce impact of the cirrus cloud is discussed in Sec. 4.3. We hope those discussions are sufficient for the readers to follow our work.

At the end of Sec 2.1 we added:

**"In this study, we will discuss the retrieval uncertainty and performance in aerosol properties, ocean surface wind speed, and Chl a in the ocean, as well as water leaving signals based on the retrieval parameters. The water leaving signal refers to the remote sensing reflectance (Rrs), which is the ratio of the upwelling water leaving radiance and the downwelling solar irradiance just above the ocean surface (Mobley 2022). Rrs can be estimated through the atmospheric correction process which removes the contribution from the atmosphere and ocean surface from the total measurements at the sensor and additional BRDF correction to reduce the dependency on the solar and viewing directions. Both atmospheric and BRDF corrections with their associated uncertainties are implemented using neural networks as discussed in Gao et al., (2021a, b) and followed by this study."**

---

## Author Comment (AC2)

We appreciate the effort and time the reviewer has invested in reviewing our manuscript. We are grateful for the constructive feedback, which have improved the quality of our research. Please find our response below with revision details.

This study by Gao et al. conducts the impact and estimation of the angular uncertainty correlation for multi-angle polarimetric remote sensing of aerosols and ocean color, through the development of various methods integrated into a practical framework. Theoretical and real retrieval uncertainties are derived based on error propagation and comparison of retrieved and true values, respectively. Overall, the methods used in this work are solid and important for the community, particularly, lots of previous studies neglected or simplified the angular uncertainty correlation for multi-angle retrieval. Also, the manuscript is well organized and presented. I have only a few minor concerns before it could be accepted by AMT.

Thanks for the positive comments.

- Eq.6, For the integrity of the article, I suggest authors specify the value of each element of the a priori error matrix, i.e, Sa, though details have been mentioned in Gao et al. 2022.

  Thank you for the suggestion. The value in the a priori error matrix is added to appendix Table A1.

  Original:

    In this study, each retrieval parameter can only vary in a limited range as shown in Table A1, which imposes an implicit a priori constraint on the retrieval parameters.

  Revised:

    In this study, each retrieval parameter can only vary in a limited range, which imposes an implicit a priori constraint on the retrieval parameters. **Both the parameter ranges and a priori values are listed in Table A1.**

- P6L11, Also, it is better to specify the equation on the calculation of theoretical uncertainty of variables which are not retrieved parameters directly but related to the state vector, e.g., remote sensing reflectance, Rrs.

  I believe the review refers to P7L11. We made the following revisions:

  Original:

    The uncertainties of variables which are a function of the retrieval parameters can also be derived from S and their derivatives. Due to the large number of retrieval parameters used in the retrieval, the evaluation of the retrieval uncertainties can be time consuming. The speed to compute uncertainties is improved using

automatic differentiations based on neural network forward models (Gao et al., 2022).

Addition:

> **… For example, the uncertainty of remote sensing reflectance (Rrs) can be derived using the automatic differentiation applied on the neural networks for BRDF correction and atmospheric correction components (Gao et al., 2022, Appendix A: Speed improvement using automatic differentiation).**

P9L5, please correct the correlation angle and correlation parameter as, θc = 60∘ (r = 0.983)

Corrected.

We also identified a similar typo in P20L5, which are also corrected: r = 0.0992 is corrected as r = 0.992.

- Fig.2, what are three sets of error examples?

We generated a total of 1000 cases, this is three random sets of error examples. It is used to demonstrate the longer range of correlations. Added more discussions:

Original:

> "…Three sets of error examples are shown in different colors."

Revised:

> **"A total of 1000 set of errors are generated and added to the simulation data.** Three error sets are shown as examples in different colors."

- P12L15, Table3.2 -> Table 3

Corrected

- P13L3, Table -> Table 3

Corrected.

- P14L5, Give -> Given.

Corrected

- P14L22, how did authors explain why the real retrieval uncertainty increases until θc reaching 10∘ to 20∘ for C4?

This is an interesting observation. Knobelspiesse et al 2012 also observe a similar behavior on RSP data. We added more discussions here:

Original:

> When the correct correlation angle is considered in the retrieval cost function (C4), the real retrieval uncertainty increases until $\theta c$ reaching 10º to 20º and then slightly decreases at higher correlation angles, for most retrieval parameters.

Addition:
> ... **Similar behavior of the information content has been reported by Knobelspiesse et al. (2012) on the study of error correlation in RSP measurements. To understand how the correlation influences retrieval accuracy, we further analyze its impacts on the eigenvalues of the error covariance matrix with details discussed in Appendix B**.

More discussions (in the Appendix B: Eigenvector decomposition in error covariance matrix) are here:
> "…As shown in Fig. B2 , when the correlation strength is strong the SIC is increasing with theoretical retrieval uncertainties estimated from error propagation such as for fine mode refractive index, wind speed and Chl a decreasing. However, when the correlation is relatively weak, SIC decreases and uncertainties increase with the correlations. The correlation angles with maximum uncertainties are different with different retrieval parameters, but generally fall within 30 degrees. This behavior may relate how the measurement uncertainties are mapped to the retrieval parameters space through Jacobian matrix. Similar behavior of the SIC has been reported by (Knobelspiesse et al., 2012) for the RSP measurement."

- P16L12, it is confusing why the retrieved wind speed indicates a larger uncertainty. Did the authors conduct the retrieval in the sun glint condition?

Thank you for the question. Since the NN used in this manuscript do not include the large magnitude glint signal, we have removed the glint angles in the retrievals (Fig 4), which contribute to a larger wind speed uncertainty. New NN including sunglint has been developed in our recent study and will be discussed in future works.

Original:

> "As shown in Fig. 8, the ratio is close to 1 for most parameters for $\theta_c = 10º$, with slightly larger impacts for effective variance for both fine and coarse mode as well as for wind speed uncertainty."

Addition:

> "... **Note that sunglint has been removed in the retrieval as shown in Fig 4, which may contribute to a larger wind speed retrieval uncertainty.**"

- Section 4, I suggest authors make a short discussion about why the reflectance at 670 nm inclines to have an over-fitting issue.

Thank you for the suggestion. We believe that 670nm band is the major contributor to the overfitting as shown in the overall cost function (Fig 10). This is likely due to the large number of angles at 670 band (60 angles) comparing with other bands (10 angles). Therefore, the 670 band is more sensitive to the angular correlation. As discussed previously, larger correlation may lead to smoother angular variation and the retrieval algorithm may fit those smoother variations as the real signal, which lead to a smaller residuals and therefore strong overfitting. We have discussions as follows in the manuscript:

> "The 670 nm band shows the largest reduction, where a ratio of 0.025 is observed for $\theta^c = 10^o$, reaching 0.015 for $\theta^c = 120^o$ , which may be due to the large number of angles (60) in the 670nm bands versus the other bands (10 or 20). "

- P20L3, bans - > bands

Corrected

- Figure 12, I suggest using different colors to indicate 670nm and 870nm.

Thanks for the suggestion. Two different colors for 670 (red) and 870 (brown) are used in this study. We agree that these two colors can be difficult to distinguish. To provide more information for this plot, the brown and red color symbols are quite close to each other, and the red color symbol is mostly lower than the brown color one when they are different.

- In conclusion, I suggest authors discuss the promising of those methods used in coastal water retrieval.

Thank you for the suggestion. We revised following discussions:

Original:

> "-Future retrieval algorithm development
> The pixel-wise theoretical uncertainties achieve a reasonably good performance to represent real retrievals when no correlation is presented. Their performances on various retrieved geophysical properties are quantified by comparing with the real retrieval errors. The difference grows bigger when the angular correlation is

stronger, which suggests convergence to local minima and indicates that more development is needed to improve the retrieval optimization."

Addition:

"… **For the development of future algorithms with more retrieval parameters, such as aerosols with more complex shape and absorption properties, and coastal waters with more complex bio-optical properties, a better characterized error model, such as the one considering angular or spectral correlations, will be helpful to identify information useful for the retrievals, and therefore improve retrieval performance and uncertainty assessment."**